# NoBOOM: Chemical Process Datasets for Industrial Anomaly Detection

**Dennis Wagner**[1][*]    **Fabian Hartung**[2][*]    **Justus Arweiler**[1][*]
**Aparna Muraleedharan**[3][*]    **Indra Jungjohann**[1][*]    **Arjun Nair**[1]
**Steffen Reithermann**[1]    **Ralf Schulz**[4]    **Michael Bortz**[5]
**Daniel Neider**[6]    **Heike Leitte**[1]    **Joachim Pfeffinger**[2]
**Sophie Fellenz**[1]    **Stephan Mandt**[7]    **Fabian Jirasek**[1]
**Jakob Burger**[3]    **Hans Hasse**[1]    **Torsten Katz**[2]
**Marius Kloft**[1]

[1]RPTU Kaiserslautern    [2]BASF SE Ludwigshafen    [3]TUM Straubing
[4]RPTU Landau    [5]Fraunhofer ITWM    [6]TU Dortmund    [7] UC Irvine

```
{wagnerd,justus.arweiler,indra.jungjohann,naira,
   r.schulz,fabian.jirasek,hans.hasse}@rptu.de
       {fabian.hartung,joachim.pfeffinger,
            torsten.katz}@basf.com
     {aparna.muraleedharan,burger}@tum.de
         {reithermann}@informatik.uni-kl.de
         {michael.bortz}@itwm.fraunhofer.de
              {neider}@mpi-sws.org
      {leitte,fellenz,kloft}@cs.uni-kl.de
               {mandt}@uci.edu
```

## Abstract

Monitoring chemical processes is essential to prevent catastrophic failures, optimize costs and profits, and ensure the safety of employees and the environment. A key component of modern monitoring systems is the automated detection of anomalies in sensor data over time, called time series, enabling partial automation of plant operation and adding additional layers of supervision to crucial components. The development of anomaly detection methods in this domain is challenging, since real chemical process data is usually proprietary, and simulated data is generally not a sufficient replacement. In this paper, we present NoBOOM, the first collection of datasets for anomaly detection in real-world chemical process data, including labeled data from a running process at our industry partner BASF SE — one of the world's leading chemical companies — and several chemical processes run in laboratory-scale and pilot-scale plants. While we are not able to share every detail about the industrial process, for the laboratory- and pilot-scale plants, we provide comprehensive information on plant configuration, process operation, and, in particular, anomaly events, enabling a differentiated analysis of anomaly detection methods. To demonstrate the complexity of the benchmark, we analyze the data with regard to common issues of time-series anomaly detection (TSAD) benchmarks, including potential triviality and bias.

**Code**: https://github.com/wagner-d/noboom

**Dataset**: https://www.kaggle.com/datasets/faebs94/noboom-anomaly-detection-in-chemical-processes[1]

---

[*]These authors contributed equally to this work.

[1]We provide the data via DOI, Kaggle, and a private server. For details and links, please refer to appendix A.

# 1 Introduction

Chemical manufacturing is a cornerstone of modern society, supplying essential components for pharmaceuticals, polymers, fuels, and consumer products. Chemical plants operate under tight safety, economic, and environmental constraints, where even small anomalies can rapidly escalate into catastrophic events [20, 28, 32] that threaten lives, endanger the environment, and cause huge economic losses. It is of paramount importance to detect anomalies as early and precisely as possible so that their causes can be addressed in time to prevent escalation.

Chemical plants produce much more data than a human operator could consistently monitor. Consequently, most modern plants rely on control systems to assist human operators in monitoring the ongoing process and detecting anomalies. Most control systems rely on simple anomaly detection methods and do not use modern developments in Machine Learning (ML). This gap is largely due to the lack of publicly available data to develop ML methods and to explore and test new ideas in a safe environment. Publishing industrial process data is laborious and, when done by companies, risks exposing sensitive operational details to competitors. Furthermore, industrial processes are usually run at single optimized operation points, and failures are actively avoided, making the available in-house data rather uninformative and unsuitable for developing powerful anomaly detection methods. Consequently, ML-based anomaly detection methods are often not a viable option for companies.

For decades, the Tennessee Eastman Process (TEP) dataset [8] has been the primary publicly available source of chemical process data. However, it contains only simulated data from a hypothetical process, which is too well-behaved to capture the complexity, noise characteristics, and operational variability of real chemical process data. To evaluate anomaly detection methods on the TEP [13], anomalies are artificially induced by adjusting the simulation [26]. Because of the lack of real chemical process data and the simplification of the TEP, it is still an open question how these results translate to real chemical processes.

To anchor anomaly detection research in a widely applicable context, we focus on *distillation*—one of the most fundamental separation processes in chemical engineering. Distillation is scalable, operates in both batch and continuous modes, and is governed by well-understood physical principles. Its ubiquity and structure make it an ideal candidate for benchmarking anomaly detection methods. Yet, no public dataset currently provides real sensor data from an operating distillation process with labeled anomalies.

Creating realistic datasets for anomaly detection in chemical processes, however, is extremely challenging for two main reasons. First, collecting real-world data is resource-intensive: processes must run for long periods under expert supervision, and inducing meaningful anomalies without compromising safety is difficult. Sensors must be carefully placed to generate rich multivariate time-series data that sufficiently captures the process and the anomalies. Second, even when data is available, designing a meaningful benchmark is difficult. Many existing TSAD datasets use oversimplified signals, trivial anomalies, or imprecise labels—failing to capture the complexity of real-world industrial behavior. Additionally, standard evaluation metrics are insufficiently aligned with industrial requirements often rewarding behaviors that contradict practical needs. As a result, progress in developing robust methods has been slow.

To fill this gap, we introduce **NoBOOM**, the first collection of real-world chemical process datasets for multivariate time-series anomaly detection. Our benchmark integrates both industrial and academic sources: labeled data from a large-scale industrial process at our industry partner BASF SE and from multiple laboratory-scale plants. Where possible, we provide extensive documentation on configuration, operational conditions, and anomaly events and phases, facilitating robust and reproducible evaluation of detection methods.

**The main contributions of this paper are:**

- We introduce **NoBoom**—a diverse suite of six real-world multivariate time-series datasets for anomaly detection in operating chemical plants, *including the first publicly available dataset of its kind derived from a major industrial partner*.
- We introduce a new evaluation protocol that leverages richer label information to enable more nuanced assessments of TSAD methods.
- We assess the data complexity through a set of simple baselines.

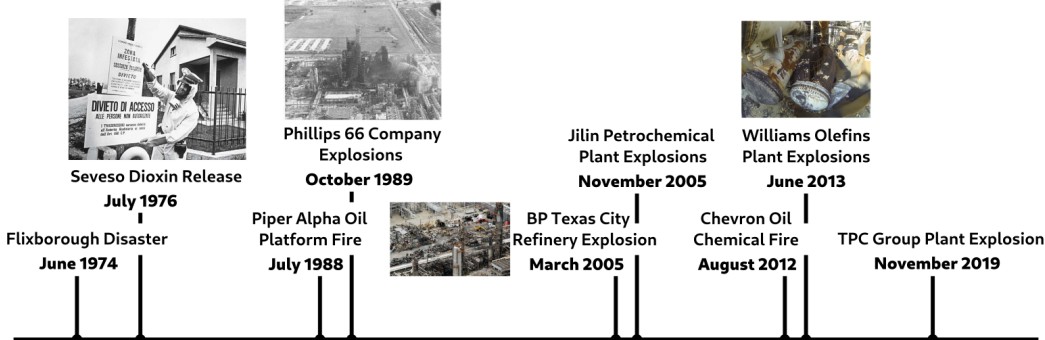

Figure 1: Timeline of major chemical plant accidents, from the Seveso dioxin release (July 1976) to the TPC Group explosion (November 2019). The figure highlights key incidents caused by preventable equipment failures or safety lapses and shows their dates, types, and locations to emphasize how recurring issues in maintenance, human decisions, and process safety have led to major industrial disasters.

## 2 BOOM: The Risk of Industrial Anomalies

Anomalies in chemical plants can lead to the unexpected release of hazardous material or even explosions. Without immediate intervention or preventive measures, such deviations can escalate into dangerous, sometimes fatal, incidents (see Figure 1). While major accidents, especially those involving fatalities, are usually investigated thoroughly and publicly reported, many smaller incidents and near misses remain inaccessible to the public.

Figure 2 illustrates sensor data from a laboratory distillation plant, where timely intervention by an attentive operator prevented a catastrophic failure. A malfunctioning cooling system caused the plant to empty unexpectedly. Within minutes, excessive heat accumulated inside the apparatus, damaging the reboiler vessel and wiring throughout the setup. By chance, the operator noticed smoke escaping from the plant and managed to cut the power, just in time to prevent a major fire. The incident caused equipment damage in the thousands but no injuries; a fire, however, would have resulted in millions and endangered many people.

This example demonstrates that anomalies in chemical processes can be extremely dangerous. While such events can never be fully avoided, their risks can be mitigated through precise, robust, and reliable anomaly detection methods. To develop such methods, we need realistic and challenging benchmark datasets.

## 3 The NoBOOM Datasets

The NoBOOM benchmark contains six datasets, each comprising both fault-free and anomalous data from real operating plants. Distillation is the most widely used unit operation, accounting for about 90–95% of separations in the chemical industry [6, 21]. Its applications span petroleum refineries, chemical production, food processing [29], and pharmaceutical manufacturing [6]. The process exploits differences in volatility by heating a mixture in a reboiler: volatile components vaporize and rise, while heavier ones remain in the liquid. The vapor is condensed, partially fed back into the distillation column as reflux, and the remainder is extracted as a product rich in more volatile components.

To construct a representative benchmark, we conducted distillation experiments across different operation modes and plant configurations. We collected five datasets from two laboratory plants that allow controlled injection of artificial anomalies, while one dataset originates from a multi-stage industrial process operating under real conditions. For artificially induced anomalies, we provide labels that distinguish three different phases, reflecting the injection of the cause of the anomaly, the effect, and the recovery after removal of the anomaly cause from the system. The distillation-

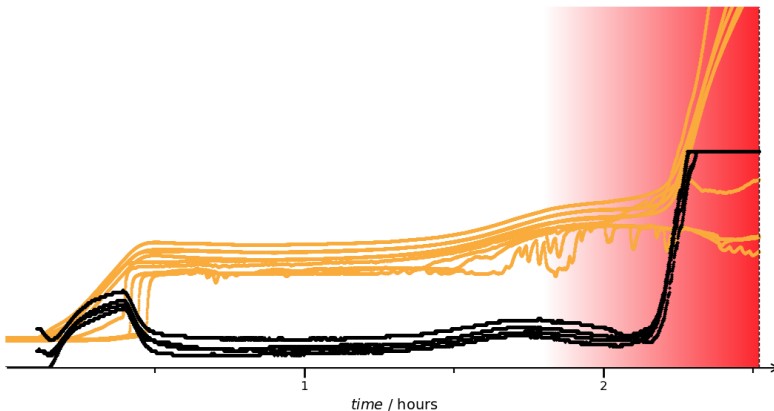

Figure 2: Critical temperature escalation in a batch-distillation process, with temperatures rising sharply over time (red-shaded area). Reliable anomaly detection is essential to prevent such hazardous developments in chemical plants. Shown are process temperatures (black) and heater actuator signals (orange) during a near-miss incident caused by a malfunctioning cooling system. About two hours into the process, uncontrolled heat buildup began and escalated rapidly, as highlighted by the red region. The plant was shut down roughly 30 minutes later, narrowly avoiding severe damage.

experiment datasets include machine-readable, ontology-based metadata [39] describing anomaly causes and the sensors that observed them.[2]

## 3.1 The Laboratory Distillation Plants

Chemical plants can generally be operated in two modes: batch and continuous. In batch mode, a certain amount of the initial mixture is fed into the plant. Then, the process is started by heating the mixture until it starts to boil. During batch distillation, the product is drawn successively from the plant. Thus, the state of the process is ever changing, with varying temperatures and mixture compositions, making the process inherently dynamic and fault detection challenging. In continuous mode, the plant is fed continuously, reaching a steady state where sensor readings fluctuate minimally, while both condensated vapor from the top of the column and liquid from the reboiler vessel at the bottom of the column are withdrawn. The plant is usually kept in a steady state of operation for long periods of time. Both modes of operation meet specific needs and are therefore widely used in industry.

### 3.1.1 The Batch-Distillation Plant

We collected two datasets in the laboratory-scale batch-distillation plant comprising measurements from 18 sensors for temperature, pressure, fluxes, and levels[3]. We provide detailed information on setup and operation, as well as actuator data of the plant in accompanying work [3], making all sensor data fully explainable and traceable. The data is sampled at 1 Hz to capture the fast-changing dynamics.

Anomalies arose naturally in the process, or were induced artificially. Thereby, time frame and frequency of anomaly induction did not follow a strict scheme to reflect the lack of a-priori knowledge on anomalies in real chemical processes. The nature of the introduced anomalies is inspired by faults and errors prevalent in accidents in industrial plants [20].

We excluded startup and shutdown phases of the process from the datasets, as their behavior differs significantly from regular operation. The datasets cover two ternary systems: acetone + methanol + 1-butanol (abm) and 1-butanol + 2-propanol + water (bpw). Sensor readings are provided in raw form, with all modifications and processing steps documented in the accompanying work mentioned above.

---

[2]We provide further information on the anomaly metadata in Appendix B.

[3]We provide further information on each sensor in the batch-distillation plant in Appendix C

### 3.1.2 The Continuous Distillation Plant

We collected three datasets from a pilot-scale continuous distillation plant with a processing capacity of 5 tonnes per year. Data was sampled every 30 seconds, sufficient to capture relevant dynamics. The plant is equipped with a multitude of sensors (up to 34) and actuators, depending on the scenario and plant configuration. [4]. Detailed information on setup, operation, and actuator data is provided in accompanying work [24].

The datasets reflect three distinct configurations: (i) a single column distilling water (`wat`), (ii) two connected columns separating n-butanol + water (`but`), and (iii) a reactive system involving water + formaldehyde + polyoxymethylene ethers (OME) + methanol (`ome`) [9]. Experiments with water span approximately 30 cumulative days. Experiments with n-butanol and water generally last about 10 hours, with some extending overnight. Experiments of the third type also typically ran for around 10 hours, occasionally spanning overnight periods. Anomalies occurred both naturally and through artificial injection, aligned with fault scenarios reported by Kister [20]. The experiments with n-butanol + water and OME result in a higher anomaly density due to their heteroazeotropic and reactive nature, as well as the increased process complexity. These configurations reflect realistic industrial conditions, providing a more rigorous testbed than datasets containing isolated, rare faults.

In addition to their technical complexity, the studied chemical systems have practical relevance: n-butanol is a widely studied biofuel with industrial applications [17, 14, 31], and OME is a promising alternative to fossil fuels [4]. The dataset thus provides not only technical but also practical value for industrial anomaly detection.

## 3.2 The Continuous Multi-Stage Industrial Plant

The industrial dataset (`ind`) was provided by our partner BASF SE and consists of long-term sensor data from a confidential, multi-stage production process. It was collected over several months of regular plant operation, reflecting a real-world industrial environment under typical business constraints. This long-term, non-experimental setting provides valuable insight into operational variability and naturally occurring anomalies, beyond short-term or lab-based datasets. In total, the dataset comprises measurements from *244 distinct sensors*, covering flow rate (F), liquid level (L), pressure (P), and temperature (T). Explicit measurement units and sampling frequencies are omitted due to confidentiality, sensor types are encoded in the feature names.

Anomalies in the dataset occurred naturally during production and were labeled retrospectively by domain experts from the partner organization. These labels are based on observed effects in the time-series data and corroborating plant operation records. The anomalies have different causes—including sensor faults and process disturbances—mirroring the complexity and heterogeneity of real-world industrial failures. To preserve confidentiality, the order of sensors within each type has been randomized, and no metadata about specific anomaly events are disclosed. However, the time-series data itself remains unaltered and fully authentic.

The dataset spans eight uninterrupted production runs across several months. Since anomaly timing was not controlled experimentally, we selected the longest segments of stable operation within each run as a potential normal-data subset for semi-supervised training. To prevent temporal leakage, the process manager of the industrial plant provided suggestions for reasonable time frames in which most long-term effects should be resolved. These initial estimates were checked by other in-house domain experts, including the plant operator. We conservatively extended these suggested intervals by doubling the initial estimate. We also excluded startup and shutdown phases, as their transient dynamics differ significantly from steady-state behavior.

## 3.3 Dataset Structure and Labeling

The NoBOOM benchmark provides a suite of real-world multivariate time-series datasets designed to evaluate anomaly detection methods in chemical process monitoring. Each dataset consists of labeled sensor readings from industrial or academic plants, covering both normal and faulty operations. We now formalize the structure of these datasets and the labeling conventions used.

---

[4]We provide further information on each sensor in the continuous distillation plant in Appendix D

Table 1: NoBOOM contains six high-dimensional datasets of varying size and complexity, from large-scale industrial data with millions of time points and hundreds of sensors to laboratory-scale experimental data with a higher anomaly density.

| mode | system | | features | time series | time steps | anomalies (# / %) |
|------|--------|------|----------|-------------|------------|-------------------|
| batch | bpw | train | 18 | 28 | 189444 | - |
| | | test | | 63 | 395712 | 77 / 21% |
| | abm | train | | 8 | 30216 | - |
| | | test | | 16 | 69558 | 13 / 24% |
| cont | wat | train | 20 | 15 | 57284 | - |
| | | test | | 11 | 37250 | 20 / 25% |
| | but | train | 34 | 7 | 2390 | - |
| | | test | | 8 | 4082 | 24 / 41% |
| | ome | train | 20 | 5 | 2447 | - |
| | | test | | 3 | 986 | 4 / 36% |
| | ind | train | 244 | 8 | 215841 | - |
| | | test | | 16 | 1842436 | 361 / 17 % |

A time series is a finite sequence $x\colon [n] \to \mathbb{R}^d$ for some $d \in \mathbb{N}$. Each sample is labeled with an integer: typically 0 (normal) or 1 (anomalous). For induced anomalies, we provide a three-phase label scheme:

1: Cause of anomaly injected; no visible effects

2: Anomaly present and recognized by operator

3: Cause of anomaly removed; effects still visible

For binary TSAD tasks, we collapse all non-zero labels into a single "anomalous" class. Labels thus form a 1D time series aligned with the inputs. Each dataset contains semantically related time series and labels. Fault-free sequences are used for training; anomalous ones for testing. Two datasets contain over 200k time steps—suitable for deep methods—while two are smaller (<3k), ideal for shallow models. For a detailed summary, see Table 1. Though the anomaly density may conflict with TSAD assumptions [33], it reflects real plant behavior and the diversity of possible anomalies. All datasets except the industry one include machine-readable metadata on anomaly causes.

Together, these datasets form the foundation of the NoBOOM benchmark, which defines a realistic anomaly detection task and an evaluation protocol, as detailed in the next section.

## 4 The NoBOOM Benchmark

The NoBOOM benchmark builds on the previously described datasets by defining a realistic anomaly detection task and a corresponding evaluation protocol. This section formalizes how models interact with the data and how their performance is assessed.

### 4.1 The Task and Evaluation Protocol

To facilitate competitive evaluations, we now discuss the goals of anomaly detection in the NoBOOM datasets. An anomaly detection algorithm $a\colon x \mapsto \{0, 1\}$ predicts the label for the next step of a given time series $x$ of length $m \in \mathbb{N}$. By predicting the label for the next time step of every prefix in a time series, we obtain a time series of prediction of the same length as the data and the labels. Different algorithms produce different predictions. To evaluate and compare the performance of different algorithms, we generally define a metric that compares the ground-truth labels with the predictions. A good metric should produce higher values for predictions that align closer with the desired behavior of a method. Thus, we first discuss the requirements of good methods for our application.

First and foremost, detecting anomalies is the most important aspect, which is not necessarily guaranteed to be reflected by common metrics [11, 22, 18, 7]. Second, the earlier an anomaly is detected, the better. Third, false alarms—anomalous predictions where the ground-truth is normal—erode the users trust in the method and should be avoided. These requirements are provably satisfied

by the ALARM score [34][5]. The ALARM score satisfies several essential properties vital to our setting. It ranks methods that detect anomalies too early (if first predictions are earlier than any indication in the labels) below methods that detect anomalies during the actual labels. Since we can pinpoint the start of each anomaly precisely for each anomalous event, no effects should be present in the data earlier, making such predictions impossible. Conversely, the labels for the end of anomaly windows are generous to mitigate any lingering effects. The ALARM score ranks predictions of anomalies for more extended periods higher than predictions that allocate false positives elsewhere. Additionally, the ALARM score favors earlier prediction for individual anomalies. Earlier predictions provide more time to respond, which can be crucial for resolving complex causes of anomalies. The ALARM score has one tunable parameter, the false alarm tolerance. This parameter reflects how many false alarms (predicted anomalies where there are none in the data) are tolerable by the system per true alarm. Since in our setting, an operator needs to manually investigate each alarm, whether false or not, false alarms can quickly degrade trust. With enough false alarms, an anomaly detector would be decommissioned promptly. Therefore, we recommend a false alarm tolerance of two.

In addition to the main evaluation metric, we highlight several supporting metrics that provide additional insights into the performance of each method. Particularly interesting is the percentage of detected anomalies, which is captured by the event-wise recall [10]. That might include anomalies that are detected before an operator detected the anomaly or after it had already been fixed. Of particular interest to our setting is the comparison to human perception. If an anomaly detector consistently predicts anomalies first in phase 3, that anomaly detector would only raise an alarm once the anomaly's cause is removed. To do so in the first place would still require a human operator to detect and fix the anomaly, limiting the potential application of the anomaly detector. The late detection frequency (LDF) captures the fraction of anomalies detected late, i.e., the fraction of anomalies where the algorithm performed worse than the operator, but still detected at least some effects. On the other hand, we can consider anomalies detected earlier than a human operator could. Similar to late alarms, we can consider early alarms, i.e., those first raised in phase 1, before the operator notices the anomaly. The early detection frequency (EDF) captures the fraction of anomalies detected faster than the expert. Methods with a high EDF consistently outperform a human operator. While not necessary, this is a strong indicator of their integrability into automated or assisted operation. An anomaly detector that only detects anomalies late is mostly useless for practical applications. Particularly interesting for operators is the average alarm frequency (AAF) measuring the consistency of predictions. The average alarm frequency (AAF) captures the expected number of alarms for detected anomalies. Without knowledge of the true labels, each alarm might indicate a different anomaly. An operator fixing an anomaly in the plant might not be able to identify whether an alarm is redundant or caused by another problem in the plant. Although none of the supplementary metrics satisfy our requirements on their own, they can provide additional insight into each methods behavior, providing a more complete picture of its capabilities. [6] Next, we analyze the dataset to verify that NoBOOM presents a non-trivial challenge.

## 4.2    Assessing Task Difficulty in NoBOOM

A dataset is generally considered trivial if the anomaly detection task can be solved by simple methods [35] that often do not require explicit training. If such a method can solve the task presented by the dataset almost perfectly, there is no need to investigate and develop complex ML methods. To assess the inherent difficulty and complexity of the NoBOOM datasets, we evaluate a set of simple baselines that detect anomalies based on deviations from the local mean, optionally after computing the first-order difference between adjacent time steps [35]. The results, summarized in Table 7, reveal that these baselines fall significantly short of ideal behavior in all datasets. This is evident not only in their ALARM scores—which, while not directly comparable across datasets, still indicate relative underperformance—but also in key operational metrics such as Average Alarm Frequency (AAF) and Early Detection Frequency (EDF). These findings suggest that NoBOOM presents a non-trivial challenge and offers meaningful opportunities for advancing robust TSAD methods.[7]

Learning algorithms are especially adept at exploiting existing biases, such as positional biases of anomalies [35]. Figure 3 shows the distribution of relative positions of anomalies in each time series

---

[5]We provide more Details in Appendix E.

[6]We provide formal definitions of each metric in Appendix E.

[7]We provide additional experiments with more baselines in appendix F.

Table 2: This table reports the results of simple baseline methods applied to all NoBOOM datasets. The ALARM scores are normalized to the maximum range. The consistently sub-optimal performance across all metrics highlights the complexity and non-triviality of the benchmark.

| dataset | ALARM | $\text{Rec}_{event}$ | $AAF$ | $EDF$ | $LDF$ |
|---|---|---|---|---|---|
| batch bpw | 0.15 | 0.18 | 2.36 | 0.07 | 0.36 |
| batch abm | 0.80 | 0.85 | 2.55 | 0.00 | 0.00 |
| cont wat | 0.12 | 0.10 | 1.50 | 0.00 | 1.00 |
| cont but | 0.38 | 0.63 | 1.13 | 0.20 | 0.73 |
| cont ome | 0.58 | 0.75 | 1.33 | 0.00 | 1.00 |
| cont ind | 0.05 | 0.05 | 1.37 | 1.00 | 0.00 |

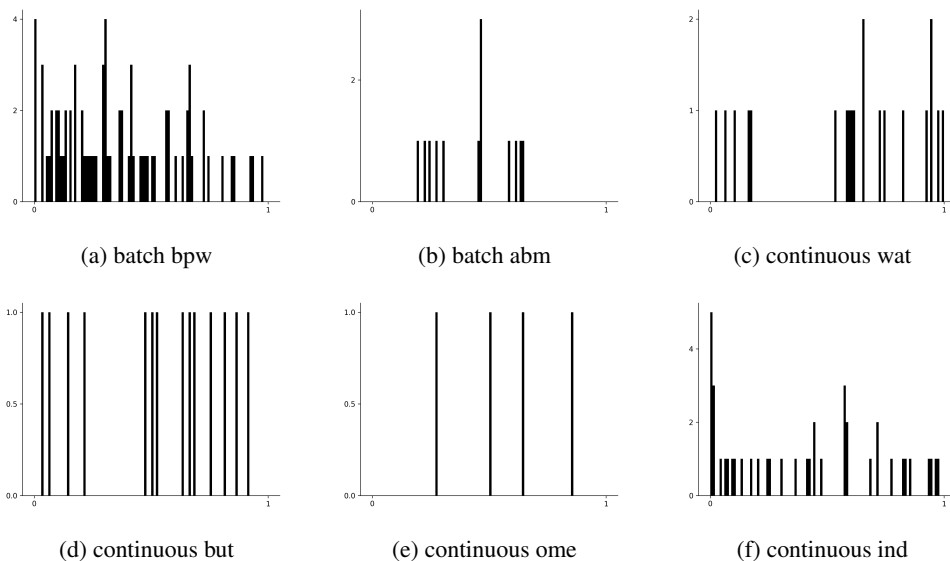

(a) batch bpw     (b) batch abm     (c) continuous wat

(d) continuous but     (e) continuous ome     (f) continuous ind

Figure 3: These plots show histogram over the relative positions of anomalies in each individual time series. The distribution of relative positions reveal no clear exploitable bias.

of all NoBOOM datasets. The results confirm no obvious exploitable bias in all datasets. Another potential issue unique to time-series data is the length of anomalies. ML algorithms usually cut a time series into manageable windows providing the context for predictions. If the length of these windows is too small, long anomalies might completely eclipse a window resulting in no normal context for predictions. Thus, overly long anomalies often distort the reported results, especially when combined with certain evaluation protocols [33]. Figure 4 shows the lengths of anomalies found in the NoBOOM datasets. Most anomalies across the entire dataset are relatively short with only a few exceptions, which can easily be isolated. Lastly, we verify the distributional stability of the normal samples between the training set and the test set. We generally assume that the normal data are sampled from the same distribution in the training and the test set. This fundamental assumption can easily be violated in real applications where anomalies can have lasting effects. In the controlled plants, we have full control over the anomalies and clear expectations for their effects, reflected in the close alignment of mean and standard deviation between training and test set for all datasets (see Figure 5).

## 5 Related Work

**Chemical Process Datasets** Industrial data are generally sparse in the public domain, since publication usually involves the risk of revealing company secrets. Therefore, the only publicly available chemical process data does not contain real process data but simulated data for a hypothetical process, the Tennessee Eastman Process (TEP) [8], which has been used to generate an anomaly detection benchmark [26]. To induce an anomaly, one of 21 predefined fault scenarios derived from the original

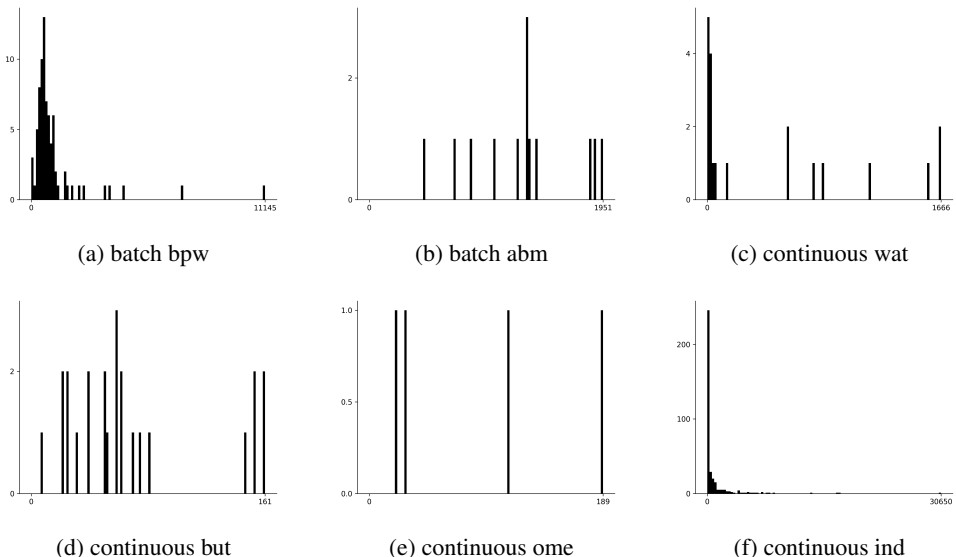

Figure 4: Histograms of anomaly lengths for each dataset. With only few notable exceptions, the vast majority of anomalies in NoBOOM are short compared to the total length of each dataset.

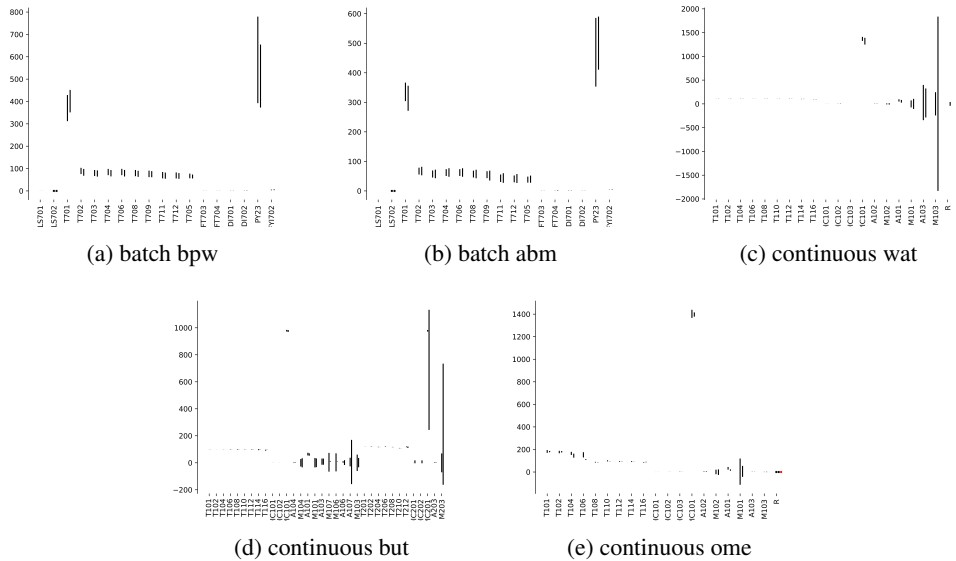

Figure 5: These plots show mean and standard deviation for each feature of every training and test set in NoBOOM next to each other, revealing no obvious significant differences in distribution that would hinder generalization. Due to the large number of features, we provide details on the remaining `ind` dataset in Appendix F.

TEP description is introduced. Anomalies can be, for example, sensor failures, valve leaks, pump faults, or changes in reaction kinetics. The data results range from step changes and sensor freezing to slow drifts. To mimic real-world behavior, anomalies start after random or fixed delayed time points. Multiple runs per fault type are provided to capture variability in anomalies [26]. Several papers have evaluated anomaly detection algorithms in TEP [13, 36, 23, 37], with inconclusive results.

**Multivariate Time-Series Anomaly Detection Datasets**  Anomaly Detection in multivariate time series has received much attention in recent years. Among the most popular datasets for comparative benchmarking are SWaT [12], WADI [2], SMAP and MSL [15], SMD [30], and Exathlon [16]. Most

timeseries datasets, both univariate [35] and multivariate [33], have been criticized for not fully capturing the complexities of the anomaly detection problem for effective benchmarking.

**Time-Series Anomaly Detection Evaluation**    Many protocols and metrics have been proposed to evaluate time series anomaly detection methods [33]. Most evaluation metrics for TSAD are point-wise or point-adjusted metrics that are naturally invariant under permutation of predictions [38, 7, 11, 19, 15, 25, 1, 19, 11, 27, 5]. Since we put great emphasis on the temporal ordering of predictions, these metrics are unsuited for our task. The metric outlined in Appendix E is based on verifiable guarantees, which largely reflect the requirements of chemical plant operation, reinforcing the strong requirements for safety and robustness.

## 6    Discussion and Conclusion

In this paper, we present NoBOOM, a collection of six datasets ranging from laboratory-scale to full industrial operation. These datasets pose realistic challenges for time-series anomaly detection across varying operational modes, process complexities, and anomaly types. Inspired by real industrial accidents [20], the controlled settings reflect critical failure modes, while the inclusion of a long-running industrial dataset marks an important step towards bridging research and practice. Together, NoBOOM offers a comprehensive benchmark for evaluating TSAD methods in chemical process monitoring.

**Limitations**    Despite its breadth, NoBOOM has several limitations. The relatively high anomaly density deviates from the classical AD setting, where anomalies are rare. While faults can be safely injected in controlled settings, extreme failures (e.g., explosions) cannot be simulated due to safety constraints. Laboratory-scale plants also cannot fully capture the complexity of industrial systems. This underscores the importance of the industrial dataset, which offers long-term production data. However, due to confidentiality restrictions, it lacks detailed annotations and documentation, limiting interpretability. These challenges reflect a broader issue: industrial data remains difficult to access and share.

**Future Directions**    The NoBOOM benchmark holds strong potential to advance ML for chemical processes. Beyond binary labels, it includes phase annotations that support the development of early-detection methods. In related work [3], additional modalities, such as tabular, audio and image, collected during distillation experiments are publicly released. This enables multimodal anomaly detection and root-cause analysis. The inclusion of actuator data further enables the exploration of causal dynamics and supports research in anomaly traceback, explainability, and process control.

**Conclusion**    In this work, we introduced NoBOOM, a benchmark suite designed to advance time-series anomaly detection (TSAD) in chemical process monitoring. By providing the first publicly available collection of real-world multivariate datasets from both laboratory-scale and industrial distillation plants, NoBOOM addresses critical gaps in the field—offering realistic, complex, and well-annotated data that reflects the true challenges of industrial anomaly detection. Through detailed metadata, phase-wise anomaly labels, and diverse operational modes, our benchmark enables both principled evaluation and the development of early, reliable, and interpretable detection methods. NoBOOM provides a foundation for the TSAD community to develop robust, practically deployable solutions that enhance the safety and efficiency of chemical production.

## Acknowledgments

The main part of this work was conducted within the DFG Research Unit FOR 5359 (ID 459419731) on Deep Learning on Sparse Chemical Process Data. SF, HH, FJ, and MK further acknowledge support by the Carl-Zeiss Foundation through the initiative Process Engineering 4.0.

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

# NeurIPS Paper Checklist

1. **Claims**

   Question: Do the main claims made in the abstract and introduction accurately reflect the paper's contributions and scope?

   Answer: [Yes]

   Justification: In the Abstract and Section 1, we announce the release of NoBOOM, a benchmark consisting of six real-world datasets from different chemical processes. This includes, for the first time, a dataset from an industrial partner, making real industrial process data publicly available. The other five datasets originate from laboratory and pilot plants that were specifically built and operated over extended periods for this purpose. In Section 3, we provide detailed descriptions of the data acquisition process, the plant setups, and the nature of the recorded anomalies. Furthermore, we introduce a new evaluation protocol for time-series anomaly detection methods with enriched label information, which we explain in Subsection 3.3 through the detailed multi-phase labeling scheme applied to the distillation datasets. Finally, in Section 4.2, we analyze the complexity and benchmarking challenges posed by these anomalies to support our final contribution.

   Guidelines:

   - The answer NA means that the abstract and introduction do not include the claims made in the paper.
   - The abstract and/or introduction should clearly state the claims made, including the contributions made in the paper and important assumptions and limitations. A No or NA answer to this question will not be perceived well by the reviewers.
   - The claims made should match theoretical and experimental results, and reflect how much the results can be expected to generalize to other settings.
   - It is fine to include aspirational goals as motivation as long as it is clear that these goals are not attained by the paper.

2. **Limitations**

   Question: Does the paper discuss the limitations of the work performed by the authors?

   Answer: [Yes]

   Justification: In Section 6, we openly discuss the limitations of our benchmark. These include the high anomaly density, which deviates from typical real-world distributions, and the safety constraints that prevent us from simulating severe incidents such as explosions. We also highlight that laboratory and pilot plants, while offering detailed control, cannot fully replicate the complexity of industrial-scale systems. Additionally, we point out that the industrial dataset, although highly valuable, comes with confidentiality restrictions that limit the availability of detailed anomaly labels and process context. This underlines the broader challenge of limited publicly accessible industrial data and emphasizes the need for more open, well-documented real-world datasets to further advance research in industrial anomaly detection.

   Guidelines:

   - The answer NA means that the paper has no limitation while the answer No means that the paper has limitations, but those are not discussed in the paper.
   - The authors are encouraged to create a separate "Limitations" section in their paper.
   - The paper should point out any strong assumptions and how robust the results are to violations of these assumptions (e.g., independence assumptions, noiseless settings, model well-specification, asymptotic approximations only holding locally). The authors should reflect on how these assumptions might be violated in practice and what the implications would be.
   - The authors should reflect on the scope of the claims made, e.g., if the approach was only tested on a few datasets or with a few runs. In general, empirical results often depend on implicit assumptions, which should be articulated.
   - The authors should reflect on the factors that influence the performance of the approach. For example, a facial recognition algorithm may perform poorly when image resolution

is low or images are taken in low lighting. Or a speech-to-text system might not be used reliably to provide closed captions for online lectures because it fails to handle technical jargon.

- The authors should discuss the computational efficiency of the proposed algorithms and how they scale with dataset size.
- If applicable, the authors should discuss possible limitations of their approach to address problems of privacy and fairness.
- While the authors might fear that complete honesty about limitations might be used by reviewers as grounds for rejection, a worse outcome might be that reviewers discover limitations that aren't acknowledged in the paper. The authors should use their best judgment and recognize that individual actions in favor of transparency play an important role in developing norms that preserve the integrity of the community. Reviewers will be specifically instructed to not penalize honesty concerning limitations.

3. **Theory assumptions and proofs**

Question: For each theoretical result, does the paper provide the full set of assumptions and a complete (and correct) proof?

Answer: [NA]

Justification: Our work focuses on the collection, description, and benchmarking of real-world datasets for time-series anomaly detection. It does not include theoretical contributions, formal assumptions, or mathematical proofs. The value of our contribution lies in the provision of high-quality empirical data and benchmarking protocols rather than in new theoretical developments.

Guidelines:

- The answer NA means that the paper does not include theoretical results.
- All the theorems, formulas, and proofs in the paper should be numbered and cross-referenced.
- All assumptions should be clearly stated or referenced in the statement of any theorems.
- The proofs can either appear in the main paper or the supplemental material, but if they appear in the supplemental material, the authors are encouraged to provide a short proof sketch to provide intuition.
- Inversely, any informal proof provided in the core of the paper should be complemented by formal proofs provided in appendix or supplemental material.
- Theorems and Lemmas that the proof relies upon should be properly referenced.

4. **Experimental result reproducibility**

Question: Does the paper fully disclose all the information needed to reproduce the main experimental results of the paper to the extent that it affects the main claims and/or conclusions of the paper (regardless of whether the code and data are provided or not)?

Answer: [Yes]

Justification: We fully disclose all relevant details required to reproduce the dataset construction, data acquisition, and labeling process for the five laboratory and pilot plant datasets. Section 3 describes the plant setups, sensor configurations, data collection procedures, and anomaly labeling schemes in detail. The released datasets include raw sensor data, metadata, and detailed labels for these experiments, making them fully reproducible and verifiable. For the industrial dataset, confidentiality restrictions prevent us from disclosing full process details or the specific nature of the anomalies. However, the raw sensor data, anonymized sensor labels, and anomaly labels are fully included in the release and can be analyzed without restrictions. While the root causes of anomalies remain undisclosed, the data itself is complete, allowing other researchers to fully reproduce our analysis on this dataset.

Guidelines:

- The answer NA means that the paper does not include experiments.
- If the paper includes experiments, a No answer to this question will not be perceived well by the reviewers: Making the paper reproducible is important, regardless of whether the code and data are provided or not.

- If the contribution is a dataset and/or model, the authors should describe the steps taken to make their results reproducible or verifiable.
- Depending on the contribution, reproducibility can be accomplished in various ways. For example, if the contribution is a novel architecture, describing the architecture fully might suffice, or if the contribution is a specific model and empirical evaluation, it may be necessary to either make it possible for others to replicate the model with the same dataset, or provide access to the model. In general. releasing code and data is often one good way to accomplish this, but reproducibility can also be provided via detailed instructions for how to replicate the results, access to a hosted model (e.g., in the case of a large language model), releasing of a model checkpoint, or other means that are appropriate to the research performed.
- While NeurIPS does not require releasing code, the conference does require all submissions to provide some reasonable avenue for reproducibility, which may depend on the nature of the contribution. For example
  - (a) If the contribution is primarily a new algorithm, the paper should make it clear how to reproduce that algorithm.
  - (b) If the contribution is primarily a new model architecture, the paper should describe the architecture clearly and fully.
  - (c) If the contribution is a new model (e.g., a large language model), then there should either be a way to access this model for reproducing the results or a way to reproduce the model (e.g., with an open-source dataset or instructions for how to construct the dataset).
  - (d) We recognize that reproducibility may be tricky in some cases, in which case authors are welcome to describe the particular way they provide for reproducibility. In the case of closed-source models, it may be that access to the model is limited in some way (e.g., to registered users), but it should be possible for other researchers to have some path to reproducing or verifying the results.

5. **Open access to data and code**

   Question: Does the paper provide open access to the data and code, with sufficient instructions to faithfully reproduce the main experimental results, as described in supplemental material?

   Answer: [Yes]

   Justification: All datasets are publicly available under the CC-BY-SA 4.0 license. We provide access to the raw sensor data, metadata, and labels for all six datasets. For the five laboratory and pilot plant datasets, we additionally provide detailed descriptions of the plant setups and anomaly causes in the paper and supplemental materials. For the industrial dataset, although certain process details and anomaly causes remain confidential, the full dataset—including anonymized sensor data and labeled anomaly intervals—is openly accessible, too. This allows researchers to use the data and reproduce our reported analyses. Usage instructions and dataset descriptions are included both in the repository and in the paper.

   Guidelines:
   - The answer NA means that paper does not include experiments requiring code.
   - Please see the NeurIPS code and data submission guidelines (`https://nips.cc/public/guides/CodeSubmissionPolicy`) for more details.
   - While we encourage the release of code and data, we understand that this might not be possible, so "No" is an acceptable answer. Papers cannot be rejected simply for not including code, unless this is central to the contribution (e.g., for a new open-source benchmark).
   - The instructions should contain the exact command and environment needed to run to reproduce the results. See the NeurIPS code and data submission guidelines (`https://nips.cc/public/guides/CodeSubmissionPolicy`) for more details.
   - The authors should provide instructions on data access and preparation, including how to access the raw data, preprocessed data, intermediate data, and generated data, etc.
   - The authors should provide scripts to reproduce all experimental results for the new proposed method and baselines. If only a subset of experiments are reproducible, they should state which ones are omitted from the script and why.

- At submission time, to preserve anonymity, the authors should release anonymized versions (if applicable).
- Providing as much information as possible in supplemental material (appended to the paper) is recommended, but including URLs to data and code is permitted.

6. **Experimental setting/details**

   Question: Does the paper specify all the training and test details (e.g., data splits, hyper-parameters, how they were chosen, type of optimizer, etc.) necessary to understand the results?

   Answer: [Yes]

   Justification: We describe the experimental setups in Section 3, covering both laboratory and industrial data collection. For each dataset, we provide information on the plant configuration, sensor types, number of time steps, sampling frequencies, and the labeling strategy. The details about training and test splits, including the distinction between normal and anomalous segments, are presented in Subsection 3.3 and Table 1. Additional sensor descriptions are provided in Appendix E. This level of detail enables readers to understand the datasets and how they are structured for anomaly detection experiments.

   Guidelines:
   - The answer NA means that the paper does not include experiments.
   - The experimental setting should be presented in the core of the paper to a level of detail that is necessary to appreciate the results and make sense of them.
   - The full details can be provided either with the code, in appendix, or as supplemental material.

7. **Experiment statistical significance**

   Question: Does the paper report error bars suitably and correctly defined or other appropriate information about the statistical significance of the experiments?

   Answer: [Yes]

   Justification: In Subsection 4.2 and Table 7, we report results of simple baseline methods across multiple runs, including averages and standard deviations for several metrics such as ALARM, event-wise recall, and average alarm frequency (AAF). These results reflect the variability in performance and provide insights into the reliability and consistency of the baselines. We clearly state which metrics are used and report statistical variations to support the assessment of dataset complexity.

   Guidelines:
   - The answer NA means that the paper does not include experiments.
   - The authors should answer "Yes" if the results are accompanied by error bars, confidence intervals, or statistical significance tests, at least for the experiments that support the main claims of the paper.
   - The factors of variability that the error bars are capturing should be clearly stated (for example, train/test split, initialization, random drawing of some parameter, or overall run with given experimental conditions).
   - The method for calculating the error bars should be explained (closed form formula, call to a library function, bootstrap, etc.)
   - The assumptions made should be given (e.g., Normally distributed errors).
   - It should be clear whether the error bar is the standard deviation or the standard error of the mean.
   - It is OK to report 1-sigma error bars, but one should state it. The authors should preferably report a 2-sigma error bar than state that they have a 96% CI, if the hypothesis of Normality of errors is not verified.
   - For asymmetric distributions, the authors should be careful not to show in tables or figures symmetric error bars that would yield results that are out of range (e.g. negative error rates).
   - If error bars are reported in tables or plots, The authors should explain in the text how they were calculated and reference the corresponding figures or tables in the text.

8. **Experiments compute resources**

   Question: For each experiment, does the paper provide sufficient information on the computer resources (type of compute workers, memory, time of execution) needed to reproduce the experiments?

   Answer: [Yes]

   Justification: The hardware and server properties are listed in Appendix F. Our work focuses on dataset creation and benchmarking with simple, lightweight baseline methods that require minimal computational resources. We do not perform large-scale model training or computationally intensive experiments. Therefore, extensive compute infrastructure is not required for the results presented in this paper.

   Guidelines:

   - The answer NA means that the paper does not include experiments.
   - The paper should indicate the type of compute workers CPU or GPU, internal cluster, or cloud provider, including relevant memory and storage.
   - The paper should provide the amount of compute required for each of the individual experimental runs as well as estimate the total compute.
   - The paper should disclose whether the full research project required more compute than the experiments reported in the paper (e.g., preliminary or failed experiments that didn't make it into the paper).

9. **Code of ethics**

   Question: Does the research conducted in the paper conform, in every respect, with the NeurIPS Code of Ethics https://neurips.cc/public/EthicsGuidelines?

   Answer: [Yes]

   Justification: We comply with the NeurIPS Code of Ethics. All data was collected in controlled laboratory environments or in collaboration with an industrial partner, without involving human subjects or personal data. We ensured that the publicly released datasets respect confidentiality agreements and do not disclose sensitive or proprietary information beyond what has been explicitly authorized for public release. Our work aims to support safety and reliability in industrial operations, which aligns with ethical principles of societal benefit.

   Guidelines:

   - The answer NA means that the authors have not reviewed the NeurIPS Code of Ethics.
   - If the authors answer No, they should explain the special circumstances that require a deviation from the Code of Ethics.
   - The authors should make sure to preserve anonymity (e.g., if there is a special consideration due to laws or regulations in their jurisdiction).

10. **Broader impacts**

    Question: Does the paper discuss both potential positive societal impacts and negative societal impacts of the work performed?

    Answer: [Yes]

    Justification: In Section 6, we discuss the potential positive impacts of NoBOOM, including enabling the development of more robust and safer anomaly detection methods for chemical processes, which could help prevent industrial accidents and improve environmental and worker safety. We do not foresee direct negative societal impacts or obvious misuse risks, as the data is technical, anonymized, and process-specific. However, we highlight the general importance of transparency and further real-world data availability to advance industrial safety research.

    Guidelines:

    - The answer NA means that there is no societal impact of the work performed.
    - If the authors answer NA or No, they should explain why their work has no societal impact or why the paper does not address societal impact.

- Examples of negative societal impacts include potential malicious or unintended uses (e.g., disinformation, generating fake profiles, surveillance), fairness considerations (e.g., deployment of technologies that could make decisions that unfairly impact specific groups), privacy considerations, and security considerations.
- The conference expects that many papers will be foundational research and not tied to particular applications, let alone deployments. However, if there is a direct path to any negative applications, the authors should point it out. For example, it is legitimate to point out that an improvement in the quality of generative models could be used to generate deepfakes for disinformation. On the other hand, it is not needed to point out that a generic algorithm for optimizing neural networks could enable people to train models that generate Deepfakes faster.
- The authors should consider possible harms that could arise when the technology is being used as intended and functioning correctly, harms that could arise when the technology is being used as intended but gives incorrect results, and harms following from (intentional or unintentional) misuse of the technology.
- If there are negative societal impacts, the authors could also discuss possible mitigation strategies (e.g., gated release of models, providing defenses in addition to attacks, mechanisms for monitoring misuse, mechanisms to monitor how a system learns from feedback over time, improving the efficiency and accessibility of ML).

11. **Safeguards**

Question: Does the paper describe safeguards that have been put in place for the responsible release of data or models that have a high risk for misuse (e.g., pretrained language models, image generators, or scraped datasets)?

Answer: [NA]

Justification: Our work does not involve the release of models or datasets with a high risk of misuse. The released data consists of anonymized sensor readings from chemical process operations without any personal, security-relevant, or ethically sensitive information. No additional safeguards beyond anonymization and confidentiality agreements with the industrial partner were necessary.

Guidelines:

- The answer NA means that the paper poses no such risks.
- Released models that have a high risk for misuse or dual-use should be released with necessary safeguards to allow for controlled use of the model, for example, by requiring that users adhere to usage guidelines or restrictions to access the model or implementing safety filters.
- Datasets that have been scraped from the Internet could pose safety risks. The authors should describe how they avoided releasing unsafe images.
- We recognize that providing effective safeguards is challenging, and many papers do not require this, but we encourage authors to take this into account and make a best faith effort.

12. **Licenses for existing assets**

Question: Are the creators or original owners of assets (e.g., code, data, models), used in the paper, properly credited and are the license and terms of use explicitly mentioned and properly respected?

Answer: [Yes]

Justification: We properly cite all external datasets and references used in our work in Section 5. The NoBOOM dataset itself is released under the CC-BY-SA 4.0 license, which is clearly specified on the dataset page. No external datasets or proprietary models are repackaged or redistributed. All other referenced benchmarks (such as TEP) are only used for comparison and are properly credited with their original sources and licenses.

Guidelines:

- The answer NA means that the paper does not use existing assets.
- The authors should cite the original paper that produced the code package or dataset.

- The authors should state which version of the asset is used and, if possible, include a URL.
- The name of the license (e.g., CC-BY 4.0) should be included for each asset.
- For scraped data from a particular source (e.g., website), the copyright and terms of service of that source should be provided.
- If assets are released, the license, copyright information, and terms of use in the package should be provided. For popular datasets, `paperswithcode.com/datasets` has curated licenses for some datasets. Their licensing guide can help determine the license of a dataset.
- For existing datasets that are re-packaged, both the original license and the license of the derived asset (if it has changed) should be provided.
- If this information is not available online, the authors are encouraged to reach out to the asset's creators.

13. **New assets**

   Question: Are new assets introduced in the paper well documented and is the documentation provided alongside the assets?

   Answer: [Yes]

   Justification: The NoBOOM datasets introduced in this paper are documented with descriptions of the data acquisition process, sensor setups, anomaly labeling schemes, and data structure in Section 3 and the supplemental materials. The datasets include metadata and usage instructions and are released under the CC-BY-SA 4.0 license. No human subject data is involved, and all data is anonymized where necessary to comply with confidentiality agreements.

   Guidelines:

   - The answer NA means that the paper does not release new assets.
   - Researchers should communicate the details of the dataset/code/model as part of their submissions via structured templates. This includes details about training, license, limitations, etc.
   - The paper should discuss whether and how consent was obtained from people whose asset is used.
   - At submission time, remember to anonymize your assets (if applicable). You can either create an anonymized URL or include an anonymized zip file.

14. **Crowdsourcing and research with human subjects**

   Question: For crowdsourcing experiments and research with human subjects, does the paper include the full text of instructions given to participants and screenshots, if applicable, as well as details about compensation (if any)?

   Answer: [NA]

   Justification: Our work does not involve any crowdsourcing or research with human subjects. All data originates from technical sensor measurements of chemical processes in laboratory and industrial environments, without human interaction or data collection involving participants.

   Guidelines:

   - The answer NA means that the paper does not involve crowdsourcing nor research with human subjects.
   - Including this information in the supplemental material is fine, but if the main contribution of the paper involves human subjects, then as much detail as possible should be included in the main paper.
   - According to the NeurIPS Code of Ethics, workers involved in data collection, curation, or other labor should be paid at least the minimum wage in the country of the data collector.

15. **Institutional review board (IRB) approvals or equivalent for research with human subjects**

Question: Does the paper describe potential risks incurred by study participants, whether such risks were disclosed to the subjects, and whether Institutional Review Board (IRB) approvals (or an equivalent approval/review based on the requirements of your country or institution) were obtained?

Answer: [NA]

Justification: Our research does not involve human subjects or personal data and therefore does not require IRB approval or equivalent ethical review. All data consists solely of technical process measurements from chemical plants.

Guidelines:

- The answer NA means that the paper does not involve crowdsourcing nor research with human subjects.
- Depending on the country in which research is conducted, IRB approval (or equivalent) may be required for any human subjects research. If you obtained IRB approval, you should clearly state this in the paper.
- We recognize that the procedures for this may vary significantly between institutions and locations, and we expect authors to adhere to the NeurIPS Code of Ethics and the guidelines for their institution.
- For initial submissions, do not include any information that would break anonymity (if applicable), such as the institution conducting the review.

16. **Declaration of LLM usage**

Question: Does the paper describe the usage of LLMs if it is an important, original, or non-standard component of the core methods in this research? Note that if the LLM is used only for writing, editing, or formatting purposes and does not impact the core methodology, scientific rigor, or originality of the research, the declaration is not required.

Answer: [NA]

Justification: We did not use any large language models (LLMs) as part of the core methodology, data processing, or scientific contributions of this work. LLMs were used only for occasional language refinement, which does not impact the scientific validity or originality of the research.

Guidelines:

- The answer NA means that the core method development in this research does not involve LLMs as any important, original, or non-standard components.
- Please refer to our LLM policy (https://neurips.cc/Conferences/2025/LLM) for what should or should not be described.

# A Accessing the data

We provide a digital object identifier for version 1.0.0 of the data introduced in this paper (`https://doi.org/10.26204/data/13`). This version of the data was used for all experiments presented in this paper. Note, the two optional files FeatureOverview_Batch.csv containing additional information about the features use ";" instead of "," to separate values.

In the future, we will extend the datasets with more experiments of existing processes and with additional processes. We provide all major and minor versions of all datasets, and additional information on a private server (`http://data.for5359.de/`).

Additionally, we provide major versions of the data on kaggle (`https://www.kaggle.com/datasets/faebs94/noboom-anomaly-detection-in-chemical-processes`). Note, that this version contains all individual datasets and the auto-generated croissant file will also include all datasets.

# B Metadata of Anomalies

To create anomalous process data, we deliberately introduced controlled perturbations in the operating plant. These changes interrupt normal operation and cause sensor readings to deviate from expected values. We refer to any such sensor-recorded deviation as an anomaly. In addition to the experimental data, the datasets therefore includes structured metadata describing each labeled anomaly. To provide a standardized, semantically grounded representation, these metadata are formalized using an ontology. Semantic Web Technologies (SWTs) [39], with ontologies as a core component, offer a framework for capturing and organizing such knowledge in a uniform, machine-interpretable way. A detailed description of the metadata can be found in [3].

Each perturbation, referred to here as a failure, was initiated at a defined time, maintained for a set duration, and then cleared by restoring the plant to normal operation. The database also includes a few experiments in which the failure remained active until the end of the run. In addition, unintended anomalies occurred in some experiments; both cases are described in the metadata. The following overview lists the different failure modes for the batch-distillation and the continuous distillation plants. Hereby, a failure alters the normal function of the system and manifests as a specific failure mode.

## B.1 Metadata for the Batch-Distillation Plant

The faults introduced during experiments at the batch distillation plant can be grouped as follows: (i) setpoint changes of actuators (e.g., heaters, thermostats, pumps, or automatic valves), (ii) compromised sensor data (e.g., added noise, drift, or flatline), and (iii) addition to or removal of substances from the plant (e.g., foaming agents or nitrogen). Details about failure modes and their affected plant components are summarized in Table 3.

Table 3: Overview of induced failure modes and their affected components for the batch-distillation plant.[8]

| Failure Mode | Affected Component |
|---|---|
| Main heat input to reboiler vessel (V001) | H701 |
| Increased/Reduced heat input to upper section of reboiler vessel (V001) | H702 |
| Increased/Reduced heat input to column sections 1–3 (C001–C003) | H704, H706, H708 |
| Increased/Reduced condenser cooling capacity (HE001, HE002) | TCU1 |
| Increased/Reduced cooling-water supply | AV716 |
| Increased/Decreased vacuum line throttling | TV1 |
| Increased/Decreased reflux ratio | P701, P702 |
| Decreased liquid level in buffer vessel (V002) | PDI702 |
| Leaking into system (inert gas ingress, air ingress, condensed distillate) | HV001–HV004, HV009 |
| Leaking out of system (reflux egress) | HV004 |
| Compromised sensor output (drift, noise, flatline) | PDI701, PDI702, T703, T705, T709, T711, T712 |
| Contamination with foaming agent | V001 |
| Reboiler residue loss (material outflow) | HV009 |

---

[8]Further details will be provided in the accompanying work [3].

## B.2 Metadata for the Continuous Distillation Plant

Details about failure modes for the continuous distillation plant are summarized in Table 4.

Table 4: Overview of induced failure modes and its affected components for the continuous distillation plant.[9]

| Failure Mode | Affected Component |
|---|---|
| Bottoms pipeline clogging | M103, A103 |
| Feed pipeline clogging | M101, A101 |
| Reduced liquid in bottom vessel (B103) | M103, A103 |
| Reduced liquid in distillate vessel (B102) | M102, A102 |
| System mass balance unsatisfied | M101, A101 |
| | PDIC101, PDIC201, PIC101, PIC201 |
| Unreliable noisy sensor output | T101, T108, T110, T112, T114 |
| | T208, T210, T212 |
| Unstable head pressure sensor | PIC101 |
| Unstable level in reboiler | PDIC101, PDIC201, M101 |
| Unstable temperature sensor | T106, T108 |

---

[9]Further details will be provided in the accompanying work [24].

# C Description of Sensors in the Batch-Distillation Plant

A network of sensors, each identified by its data-stream name, ensures continuous monitoring of critical points in the batch distillation plant. Temperature sensors measure the temperature of the secondary heating jackets (T702, T704, T706, T708) that temper the column walls, as well as the primary heating jacket (T701) that supplies the main heat for evaporation. Additional temperature sensors detect the fluid temperature in the reboiler vessel V001 (T703) and at successive points along the column (T705, T709, T711, T712). Flow meters quantify the cooling-water feed (FYI702), the withdrawn distillate (FT703), and the reflux stream returned to the top of the column (FT704). Pressure is measured at the column head (PY23). Two differential-pressure sensors determine the liquid level in buffer vessel V002 (PDI702) and the pressure drop across the column (PDI701). Two level switches (LS701, LS702) provide binary wet/dry indications for the reboiler and buffer vessels, respectively. Table 5 summarizes each sensor's data-stream name, measured physical phenomenon and unit, a time-series property indicating whether the value is expected to vary during operation or remain approximately stationary, and the specific part of the plant monitored by the sensor. A more detailed discussion of the plant setup and its instrumentation will be provided in the accompanying work [3].

Table 5: List of sensor parameters for the batch distillation plant.

| Sensor Name | Measured Phenomenon | Unit | Time-Series Property | Monitored Part of Plant |
|---|---|---|---|---|
| FT703 | Flow rate | kg h$^{-1}$ | dynamic | Distillate stream |
| FT704 | Flow rate | kg h$^{-1}$ | dynamic | Reflux stream |
| FYI702 | Flow rate | pulses per second (pps) | stationary | Cooling medium stream |
| LS701 | Level | - | stationary | Liquid presence (0 = dry, 1 = wet) in reboiler vessel V001 |
| LS702 | Level | - | stationary | Liquid presence (0 = dry, 1 = wet) in buffer vessel V002 |
| PDI701 | Pressure difference | mbar | dynamic | Pressure drop across column |
| PDI702 | Pressure difference | mbar | stationary | Liquid level in buffer vessel V002 |
| PY23 | Pressure | mbar | stationary | Top of column |
| T701 | Temperature | °C | dynamic | Main heater H701 of reboiler vessel V001 |
| T702 | Temperature | °C | dynamic | Upper heater H702 of reboiler vessel V001 |
| T706 | Temperature | °C | dynamic | Heater H706 for column section 1 |
| T704 | Temperature | °C | dynamic | Heater H704 for column section 2 |
| T708 | Temperature | °C | dynamic | Heater H708 for column section 3 |
| T703 | Temperature | °C | dynamic | Residue in reboiler vessel V001 |
| T705 | Temperature | °C | dynamic | Fluid in column section 3 |
| T709 | Temperature | °C | dynamic | Fluid beneath column section 1 |
| T711 | Temperature | °C | dynamic | Fluid between column sections 1 and 2 |
| T712 | Temperature | °C | dynamic | Fluid between column sections 2 and 3 |

# D Description of Sensors in Continuous Distillation Plant

Flow rates of feed, bottom product, distillate and intermediate streams from the decanter are measured using weighing scales (for e.g., A101, M101). Differential pressure indicators (e.g., PDIC101) are located in the reboiler and distillate buffer tanks, in order to maintain the liquid levels, and they are used according to the plant configuration and the chemical system. Pressure indicators are placed at the column head (e.g., PIC101) to measure the operating pressure within the column. Temperature sensors (e.g., T101) are strategically placed in essential points within the column: the reboiler, the base of each section, and the top of the column. Further detailed discussion of the plant, sensors, manufacturers, process configurations, instrumentation diagram, and control strategies will be provided in the accompanying work [24].

Table 6: List of sensors in the continuous distillation plant data.

| Sensor Name | Measured Phenomenon | Unit | Monitored Part of Plant |
|---|---|---|---|
| A101 | Mass | kg | Feed tank |
| A102 | Mass | kg | Distillate buffer tank B102 column 1 |
| A103 | Mass | kg | Bottom product tank B103 column 1 |
| A106 | Mass | kg | Upper phase buffer tank B106 from decanter |
| A107 | Mass | kg | Lower phase buffer tank B107 from decanter |
| A203 | Mass | kg | Bottom product tank B203 column 2 |
| M101 | Flow rate | $kg\,h^{-1}$ | Feed tank |
| M102 | Flow rate | $kg\,h^{-1}$ | Distillate buffer tank B102 column 1 |
| M103 | Flow rate | $kg\,h^{-1}$ | Bottom product tank B103 column 1 |
| M106 | Flow rate | $kg\,h^{-1}$ | Upper phase buffer tank B106 from decanter |
| M107 | Flow rate | $kg\,h^{-1}$ | Lower phase buffer tank B107 from decanter |
| M203 | Flow rate | $kg\,h^{-1}$ | Bottom product tank B203 column 2 |
| PDIC101 | Pressure difference | mbar | Reboiler column 1 |
| PDIC102 | Pressure difference | mbar | Reboiler column 1 |
| PDIC201 | Pressure difference | mbar | Reboiler column 2 |
| PDIC202 | Pressure difference | mbar | Reboiler column 2 |
| PDIC103 | Pressure difference | mbar | Distillate buffer tank B102 |
| PIC101 | Pressure | mbar | Top of column 1 |
| PIC201 | Pressure | mbar | Top of column 2 |
| R | Reflux ratio | $g\,g^{-1}$ | Reflux ratio at top of column 1 |
| T101 | Temperature | °C | Reboiler column 1 |
| T102 | Temperature | °C | Bottom of section C101 |
| T104 | Temperature | °C | Bottom of section C102 |
| T106 | Temperature | °C | Bottom of section C103 |
| T108 | Temperature | °C | Bottom of section C104 |
| T110 | Temperature | °C | Bottom of section C105 |
| T112 | Temperature | °C | Bottom of section C106 |
| T114 | Temperature | °C | Below condenser |
| T201 | Temperature | °C | Reboiler column 2 |
| T202 | Temperature | °C | Bottom of section C201 |
| T204 | Temperature | °C | Bottom of section C202 |
| T206 | Temperature | °C | Bottom of section C203 |
| T208 | Temperature | °C | Bottom of section C204 |
| T210 | Temperature | °C | Bottom of section C205 |
| T212 | Temperature | °C | Bottom of section C206 |

# E Evaluation Metrics

In this section we present the formal definitions of the metrics used for the evaluation of anomaly detection methods on NoBOOM.

The ALARM score is defined as

$$|DA(p,g)| + \frac{\sum\limits_{A \in DA(p,g)} \frac{1 + \sum\limits_{i \in A} p_A(i) \cdot 2^{-i}}{2^{|I_1(p_A)|}}}{|DA(p,g)|} - \frac{|TA(p,g)| + \frac{3}{2}|EA(p,g)| + \frac{1}{2}|LA(p,g)|}{t} - (1 - \frac{1}{|p_{g=0}^{-1}(1)|})$$

where $\sigma(x) = (1 + e^{-x})^{-1}$, $EA(s,g) = \{A \in I_1(s_I) | I \in I_{01}(g > 0) \wedge (g > 0)_A \text{ surjective}\}$, $LA(s,g) = \{A \in I_1(s_I) | I \in I_{10}(g > 0) \wedge (g > 0)_A \text{ surjective}\}$, $TA(s,g) = \{A \in I_1(s) | g_A = 0 \wedge \nexists A' \in LA(s,g) \cup EA(s,g) \colon A \subset A'\}$, and $DA(s,g) = \{A \in I_1(g > 0) | \exists A_s \in I_1(s) \colon A_s \cap A \neq \emptyset \wedge (\min A_s \in A \vee g_{[\min A_s, \min A)} = 0)\}$ with $I_v(s) = \{[l,u] \subset s^{-1}(v) \mid s_{[l,u]} = v \wedge \nexists [l,u] \subsetneq [\hat{l}, \hat{u}] \subset s^{-1}(v) \colon s_{[\hat{l},\hat{u}]} = v\}$ and $I_{uv}(s) = \{[l,b] \sqcup [b+1, u] \subset s^{-1}(\{u,v\}) | [l,b] \in I_u(s) \wedge [b+1, u] \in I_v(s)\}$.

The event-wise recall is defined as

$$Recall_{event}(g,p) = \begin{cases} \frac{|\{W \in I_1(g) \colon W \cap p^{-1}(1) \neq \emptyset\}|}{|I_1(g)|} & , |I_1(g)| > 0 \\ 0 & , |I_1(g)| = 0 \end{cases}$$

The average alarm frequency (AAF) is defined as

$$AAF(p,g) = \begin{cases} \frac{\sum\limits_{W \in I_1(g) \colon |p_W^{-1}(1)| > 0} |I_1(pw)|}{|DA(s,g)|} & , |\{W \in I_1(g) \colon |p_W^{-1}(1)| > 0\}| > 0 \\ 0 & , |\{W \in I_1(g) \colon |p_W^{-1}(1)| > 0\}| = 0 \end{cases}$$

where $DA(p,g) = \{A \in I_1(g) | \exists A_s \in I_1(s) \colon A_s \cap A \neq \emptyset \wedge (\min A_s \in A \vee g_{[\min A_s, \min A)} = 0)\}$.

The early detection frequency (EDF) is defined as

$$EDF(p,g) = \begin{cases} \frac{\sum\limits_{W \in I_1(g) \colon |p_W^{-1}(1)| > 0} 1(|g_{p_W = 1}^{-1}(1)| > 0)}{|\{W \in I_1(g) \colon |p_W^{-1}(1)| > 0\}|} & , |\{W \in I_1(g) \colon |p_W^{-1}(1)| > 0\}| > 0 \\ 0 & , |\{W \in I_1(g) \colon |p_W^{-1}(1)| > 0\}| = 0 \end{cases}$$

The late detection frequency (LDF) is defined as

$$LDF(p,g) = \begin{cases} \frac{\sum\limits_{W \in I_1(g) \colon |p_W^{-1}(1)| > 0} 1(|g_{p_W = 1}^{-1}(\{1,2\})| = 0)}{|\{W \in I_1(g) \colon |p_W^{-1}(1)| > 0\}|} & , |\{W \in I_1(g) \colon |p_W^{-1}(1)| > 0\}| > 0 \\ 0 & , |\{W \in I_1(g) \colon |p_W^{-1}(1)| > 0\}| = 0 \end{cases}$$

## F  Experimental Details

For all our experiments, we use Python 3.13[10] with PyTorch 2.7[11], and CUDA 12.8.[12]. We run all our experiments on an NVIDIA DGX Cluster of 2 CPUs, 40 Cores, and 512 GB of memory.

We evaluate simple baselines of the form

$$score(x_w) = (x_{1:w} - x_{0:w-1} > mean(x_{1:w} - x_{0:w-1}) + c \cdot std(x_{1:w} - x_{0:w-1}) + b)$$

and

$$score(x_w) = (abs(x_{1:w} - x_{0:w-1}) > mean(abs(x_{1:w} - x_{0:w-1})) + c \cdot std(abs(x_{1:w} - x_{0:w-1})) + b)$$

introduced in [35] for varying values of $w, c$, and $b$ for every time series in each dataset. $mean$ and $std$ compute mean and standard deviation over time of a time series window, and $abs$ computes the absolute value element-wise. We test with and without using the mean, $c \in \{1, 2, 3, 4\}$, on the time series and first order difference, with and without the absolute value, and for a window size in $\{5, 10, 25, 50\}$. We present the results in table 7.

Additionally, we evaluate several established shallow baselines. We evaluate k-means for $k \in \{1, 10, 25, 50\}$ and window sizes in $\{5, 10, 25, 50\}$. We evaluate EIF with 200 and 500 trees, sample sizes 256 and 512, window sizes 10 and 50, and with no and zero extension levels.

Finally, we provide results for LSTM-AE with hidden dimensions [50], [50, 50], [50, 50, 50], and [50, 50, 50, 50], window size 10 and 50, and learning rates 0.001 and 0.0001. For the LSTM-P we evaluate hidden dimensions [30, 30] and [50, 50], size of the linear layer [40] and [20, 20], window sized 10 and 50, predictions horizons 3 and 10, and learning rates 0.001 and 0.0001.

For each method we train and evaluate all parameter configurations and compute the ALARM score averaged over five seeds. For the best performing model, we compute the other metrics. To obtain the predictions from the anomaly scores, we perform a line search over the scores in each dataset.

Table 7: Evaluation of additional simple baselines. The ALARM scores are normalized to the maximum range.

| dataset | method | ALARM | $Rec_{event}$ | $AAF$ | $EDF$ | $LDF$ |
|---------|--------|-------|---------------|-------|-------|-------|
| batch bpw | kmeans | 0.13 | 0.13 | 1.20 | 0.00 | 0.40 |
|  | EIF | 0.03 | 0.04 | 10.61 | 0.22 | 0.26 |
|  | LSTM-AE | 0.13 | 0.13 | 2.04 | 0.10 | 0.30 |
|  | LSTM-P | 0.10 | 0.13 | 3.44 | 0.18 | 0.35 |
| batch abm | kmeans | 0.81 | 0.85 | 1.89 | 0.00 | 0.00 |
|  | EIF | 0.03 | 0.06 | 3.95 | 0.00 | 0.05 |
|  | LSTM-AE | 0.74 | 0.94 | 5.09 | 0.03 | 0.10 |
|  | LSTM-p | 0.67 | 0.98 | 3.83 | 0.08 | 0.06 |
| cont wat | kmeans | 0.00 | 0.00 | 0.00 | 0.00 | 0.00 |
|  | EIF | 0.10 | 0.09 | 1.60 | 0.00 | 1.00 |
|  | LSTM-AE | 0.00 | 0.00 | 0.00 | 0.00 | 0.00 |
|  | LSTM-P | 0.00 | 0.00 | 0.00 | 0.00 | 0.00 |
| cont but | kmeans | 0.06 | 0.58 | 1.03 | 0.21 | 0.79 |
|  | EIF | 0.09 | 0.10 | 3.40 | 0.37 | 0.63 |
|  | LSTM-AE | 0.10 | 0.11 | 3.27 | 0.43 | 0.57 |
|  | LSTM-P | 0.16 | 0.67 | 1.12 | 0.18 | 0.82 |
| cont ome | kmeans | 0.50 | 0.50 | 1.00 | 0.00 | 0.00 |
|  | EIF | 0.10 | 0.13 | 0.50 | 0.00 | 0.50 |
|  | LSTM-AE | 0.27 | 0.25 | 1.60 | 0.00 | 0.00 |
|  | LSTM-P | 0.26 | 0.25 | 1.80 | 0.00 | 0.00 |
| cont ind | kmeans | 0.04 | 0.10 | 1.35 | 1.00 | 0.00 |
|  | EIF | 0.01 | 0.01 | 20.04 | 1.00 | 0.00 |
|  | LSTM-AE | 0.09 | 0.10 | 2.27 | 1.00 | 0.00 |
|  | LSTM-P | 0.08 | 0.10 | 2.22 | 1.00 | 0.00 |

---

[10]https://python.org

[11]https://pytorch.org

[12]https://docs.nvidia.com/cuda/archive/12.8.0/

# G Additional Evaluations

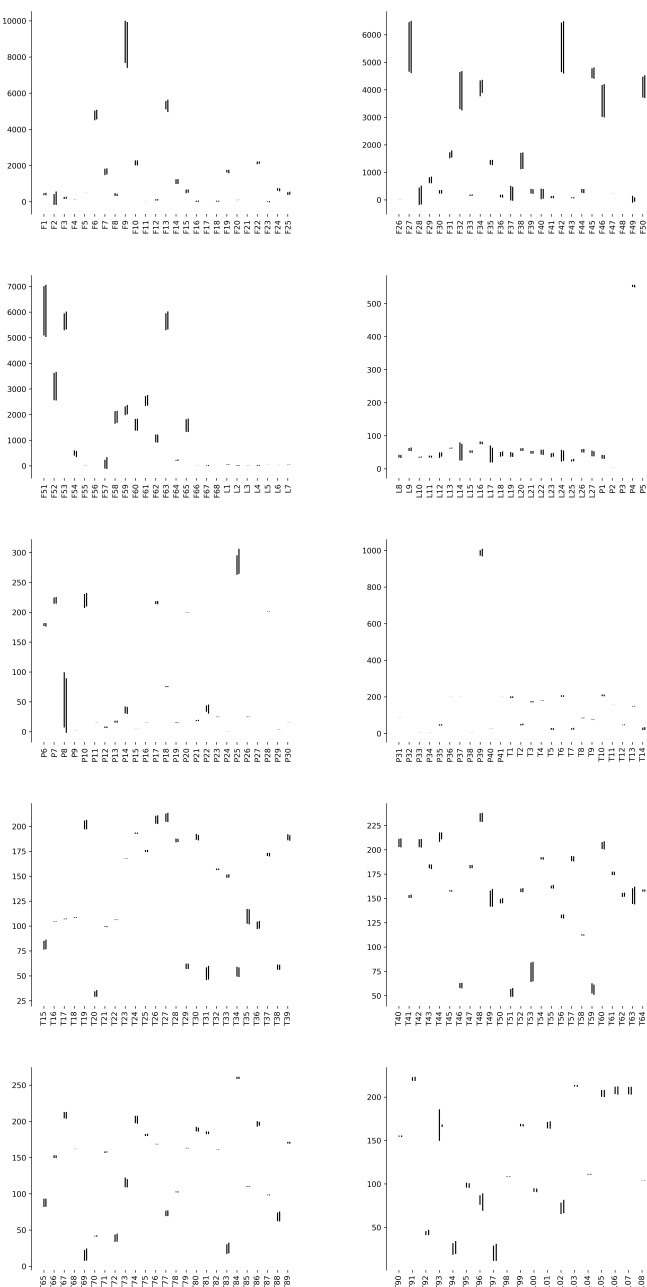

Figure 6: Distribution differences between training and test set in the industry process reveal no significant shift.

