# OpenReview forum: "NoBOOM: Chemical Process Datasets for Industrial Anomaly Detection"
_NeurIPS.cc/2025/Datasets_and_Benchmarks_Track — NeurIPS 2025 Datasets and Benchmarks Track poster_

### Official Review · Reviewer_o2DF · 2025-06-15

**Rating:** 6
**Confidence:** 2

**Summary:**

This paper presents NoBOOM, a benchmark suite comprising six real-world multivariate time-series datasets from chemical process monitoring. It includes data from both lab/pilot setups and a real industrial plant, annotated with multi-phase labels for anomaly detection. The authors propose a principled evaluation protocol and validate the dataset’s complexity using baseline methods.

**Additional Feedback:**

Even though I am not deeply familiar with evaluating papers in the benchmark track, I can see NoBOOM is indeed a highly valuable and timely benchmark that addresses a long-standing gap in industrial anomaly detection research. Its strengths—realism, diversity, careful design, and open release—greatly outweigh its limitations.

**Dataset Code Accessibility:**

Yes

**Dataset Code Comments:**

The dataset is organized clearly by experimental condition (e.g., batch vs. continuous mode), with a consistent structure for training/test splits, labels, and metadata. This modular organization makes it easy to integrate with TSAD pipelines. And the dataset is released under a permissive CC-BY-SA 4.0 license, and the documentation is sufficient to get started without the paper. This facilitates wide adoption.

**Ethical Considerations:**

No, there are no or only very minor ethics concerns

**Final Justification:**

I still think this paper has enough contribution for acceptance as its dataset are quite valuable. So I keep my score unchanged.

**Limitations Weaknesses:**

The industrial-scale dataset is extremely valuable, but the intepretebility is limited as detailed annotations and documentation are lacked.

Only basic statistical baselines are benchmarked. The paper would be stronger if it evaluated state-of-the-art deep TSAD methods (e.g., USAD, OmniAnomaly, TadGAN, Transformer-based models).

**Strengths Contributions:**

The paper presents NoBOOM, the first benchmark suite offering real sensor data from industrial and laboratory-scale chemical processes, specifically designed for multivariate time-series anomaly detection (TSAD). This contribution fills a critical gap in the field, where most existing datasets (e.g., TEP, SWaT, WADI) are either fully simulated or narrow in scope.

NoBOOM includes six distinct datasets, covering: Batch and continuous operation modes, industrial-scale processes, and diverse chemical systems (e.g., acetone + methanol + butanol; water + formaldehyde + methanol). The dataset captures both natural anomalies and artificially injected faults, offering realistic failure modes grounded in real industrial incidents.

All datasets are made publicly available under CC-BY-SA 4.0. Extensive documentation and metadata are provided for lab/pilot datasets, including sensor configurations, plant schematics, and anomaly descriptions.

---

> ### Author Rebuttal · Authors · 2025-07-31
>
> Thank you for your review and encouraging feedback. We will address your concerns individually below. Please let us know if this added explanation improves your reader's experience and makes the evaluation part more grounded.
>
> # Evaluation
> The main goal of this paper is to introduce a collection of real-world datasets for evaluating time-series anomaly detection methods. The purpose of the evaluation is to illustrate the complexity of the dataset. Shallow methods can often perform well, raising questions about whether deep-learning-based methods are necessary to solve those tasks [Keogh]. We will include more evaluations of well-established shallow baselines to strengthen our evaluation. Most modern methods are designed and tested for datasets with other statistics. Adapting and sufficiently evaluating most methods is beyond the scope of this study due to their complexity and number of hyperparameters. Furthermore, recent work [Wagner] has shown inconsistent performance of most complex models. Therefore, we will add two simple yet relevant methods [Malhotra-1, Malhotra-2] to provide a reasonable baseline for future evaluations. The following table shows the ALARM scores of each baseline method on all datasets. We will add the full evaluation with all metrics to the camera-ready version.
> | Method     | bpw   | abm   | wat   | but   | ome   | ind   |
> |------------|-------|-------|-------|-------|-------|--------|
> | threshold  | 1.87  | 1.20  | 1.31  | 2.15  | 1.79  | 2.38   |
> | kmeans     | 14.27 | 6.92  | -0.5  | 6.87  | 1.36  | 7.09   |
> | EIF        | 5.0   | 6.51  | 2.26  | 6.86  | 1.0   | 1.0    |
> | LSTM-AE    | 11.07 | **9.39** | -0.5  | 3.99  | 2.75  | **14.06** |
> | LSTM-P     | 8.13  | 3.40  | -0.5  | 1.96  | 4.0   | **8.42**  |
> | best       | 63.25 | 12.45 | 19.36 | 23.54 | 11.00 | 360.48 |
> The evaluation with shallow methods and comparatively simple deep baselines illustrates the complexity of the dataset and the need for more sophisticated methods to solve such complex tasks.
>
>
> [Keogh] https://www.dropbox.com/scl/fi/cwduv5idkwx9ci328nfpy/Problems-with-Time-Series-Anomaly-Detection.pdf?rlkey=d9mnqw4tuayyjsplu0u1t7ugg&e=2&dl=0
>
> [Wagner] Wagner, Dennis, et al. "Timesead: Benchmarking deep multivariate time-series anomaly detection." Transactions on Machine Learning Research (2023).
>
> [Malhotra-1] Malhotra, Pankaj, et al. "LSTM-based encoder-decoder for multi-sensor anomaly detection." arXiv preprint arXiv:1607.00148 (2016).
>
> [Malhotra-2] Malhotra, Pankaj, et al. "Long short term memory networks for anomaly detection in time series." Proceedings. Vol. 89. No. 9. 2015.

---

### Official Review · Reviewer_RuUy · 2025-06-29

**Rating:** 5
**Confidence:** 3

**Summary:**

The paper presents NoBOOM, a collection of datasets for anomaly detection in real-life chemical process data, including labelled data from a running process at a leading industry partner, and several chemical processes run in a laboratory-scale plant and a pilot-scale plant. The authors also proposed a new protocol to evaluate time series anomaly detection.

**Dataset Code Accessibility:**

Yes

**Ethical Considerations:**

No, there are no or only very minor ethics concerns

**Final Justification:**

I updated my score since the authors addressed all my comments regarding the evaluation protocol.

**Limitations Weaknesses:**

The authors state that one of the main contributions of the paper is the development of a new evaluation protocol. However, this protocol is barely discussed in the main text, with all technical definitions deferred to the appendix. In the appendix, only the formulae are provided, without any explanation of why the alarm score takes its specific form. In my opinion, a more thorough explanation of the score's formulation and the metrics used should be included in the main text.

**Strengths Contributions:**

Overall, the paper is well written. The datasets and benchmark are well motivated, with a clear problem in mind. The authors clearly discuss the related work and the limitations of their approach. I particularly appreciated the inclusion of the task difficulty analysis, which I found very useful for understanding the dataset.

---

> ### Author Rebuttal · Authors · 2025-07-31
>
> Thank you for your valuable comments and review. We will address your points below. With these additions, we aim to make the content more apparent to the reader. Please let us know if your concerns are addressed and appropriately resolved.
>
> # Evaluation Protocol
> The introduced evaluation protocol consists of (1) the detailed labels and (2) the metrics that can be computed on these labels. We discuss both aspects in Sections 4.1 and 3.3, respectively. The ALARM metric is rigorously introduced in another paper, which we will reference in the camera-ready version. We choose this metric as the primary evaluation metric, as it satisfies the requirements outlined in l202-l206 of Section 4.1. The novel metrics introduced in this paper are the metrics that explicitly use the additional label information, namely EDF and LDF. We provide the general intuition for each metric in Section 4.1. Still, we will gladly add detailed explanations of each metric to the Appendix and add more detail to the main paper to improve the intuition of each aspect.
> We will add the following to Section 4.1:
> > [...] LDF (Late Detection Frequency). These metrics are enabled by the phase annotations provided (Section 3.3). If an anomaly was first predicted in phase 1, it is considered "predicted early". If it was detected first in phase 3, it is considered "predicted late". An anomaly detector that only detects anomalies late is mostly useless for practical applications. In contrast, an anomaly detector that consistently detects anomalies early could set the foundation for more automation in a plant.
>
> We will add the following details to the discussion of the ALARM score:
> > The ALARM score satisfies several essential properties vital to our setting. It ranks methods that detect anomalies too early (if first predictions are earlier than any indication in the labels) below methods that detect anomalies during the actual labels. Since we can pinpoint the start of each anomaly precisely for each anomalous event, no effects should be present in the data earlier, making such predictions impossible. Conversely, the labels for the end of anomaly windows are generous to mitigate any lingering effects. The ALARM score ranks predictions of anomalies for more extended periods higher than predictions that allocate false positives elsewhere. Additionally, the ALARM score favors earlier prediction for individual anomalies. Earlier predictions provide more time for responses, which can be crucial to resolving complex causes of anomalies. Finally, the ALARM score has one tunable parameter, the "false alarm tolerance". This parameter reflects how many false alarms (predicted anomalies where there are none in the data) are tolerable by the system per true alarm. Since in our setting, an operator needs to manually investigate each alarm, whether false or not, false alarms can quickly degrade trust. With enough false alarms, an anomaly detector would be decommissioned promptly. Therefore, we recommend a default false alarm tolerance of two.
>
> We will add the following to the description of the AAF:
> > The average alarm frequency (AAF) captures the expected number of alarms for detected anomalies. Without knowledge of the true labels, each alarm might indicate a different anomaly. An operator fixing an anomaly in the plant might not be able to identify whether an alarm is redundant or caused by another problem in the plant.
>
> We will add the following to the description of the EDF and LDF:
> > Of particular interest to our setting is the comparison to human perception. If an anomaly detector consistently predicts anomalies first in phase 3, that anomaly detector would only raise an alarm once the anomaly's cause is removed. To do so in the first place would still require a human operator to detect and fix the anomaly, limiting the potential application of the anomaly detector. The late detection frequency (LDF) captures the fraction of anomalies detected late, i.e., the fraction of anomalies where the algorithm performed worse than the operator, but still detected at least some effects. On the other hand, we can consider anomalies detected earlier than a human operator could. Similar to late alarms, we can consider early alarms, i.e., those first raised in phase 1, before the operator notices the anomaly. The early detection frequency (EDF) captures the fraction of detected anomalies faster than the expert. Methods with a high EDF consistently outperform a human operator. While not necessary, this is a strong indicator of their integrability into automated or assisted operation.
>
> Additionally, we will polish the paper's main section further to highlight each component's intuition better.

---

### Official Review · Reviewer_77qs · 2025-07-03

**Rating:** 4
**Confidence:** 4

**Summary:**

The submission introduces **NoBOOM**, a benchmark suite comprising six real-world datasets for time-series anomaly detection in chemical processes, ranging from laboratory-scale to industrial-scale operations.
These datasets address the lack of publicly available, high-quality data in the chemical industry, offering detailed sensor measurements and anomaly labels that reflect realistic operational conditions.
Unlike previous synthetic benchmarks like the Tennessee Eastman Process, NoBOOM captures true process variability and complex fault dynamics, including multi-phase anomaly labeling (injection, effect, recovery).
The paper proposes a new evaluation protocol based on the **ALARM score**, which emphasizes early detection, precision, and operational consistency—key requirements for industrial safety. Through baseline evaluations and dataset complexity analysis, the authors demonstrate that NoBOOM presents a **non-trivial challenge** for modern anomaly detection methods, making it a valuable resource for advancing robust, ML-based monitoring systems in industrial settings.

**Additional Feedback:**

Please refer to the **Limitations and Weaknesses** section for detailed feedback. Here are some additional comments and questions for the authors:

1. How do you plan to address the **high anomaly density** in datasets like **abm** (28%) to ensure generalization to real-world scenarios with rare anomalies?
2. Could you elaborate on the **industrial partner’s confidentiality constraints**? Are there plans to release synthetic data or anonymized root-cause descriptions for deeper analysis?
3. Why was the **ALARM score** chosen over established metrics like F1-score or precision-recall? Could you provide a comparative analysis in Appendix D?
4. How were **hyperparameters** selected for the mean-deviation baselines? Were they tuned per dataset, and if so, what ranges were explored?
5. Could you clarify the **

**Dataset Code Accessibility:**

Yes

**Dataset Code Comments:**

Code and dataset are provided and executable.

**Ethical Considerations:**

No, there are no or only very minor ethics concerns

**Final Justification:**

Most of my initial questions, including the new feedback question, are addressed. I am still conservative about the benchmarking with baselines, but overall, it turned out to be a good dataset for the scientific domain. Raised my initial assessment.

**Limitations Weaknesses:**

### **High Anomaly Density vs. Real-World Scenarios**
The paper acknowledges that anomalies in the datasets (e.g., **ind** dataset with 17% anomaly density; __Table 1__) are denser than in real-world settings, where anomalies are typically rare. This discrepancy risks overfitting models to anomalies during training, undermining generalization. For instance, the **abm** dataset has 28% anomalies in the test set, far exceeding typical industrial failure rates [6].
So the paper should consider:
- **Providing guidelines** for handling class imbalance (e.g., stratified sampling, cost-sensitive learning).
- **Including semi-supervised training splits** with stricter constraints on normal data to mimic real-world scarcity of anomalies.

### **Confidentiality Constraints in the Industrial Dataset**
The **ind** dataset, while unique, lacks detailed metadata (e.g., sensor units, anomaly causes) due to confidentiality agreements (Section 3.2). This limits interpretability and reproducibility. For example:
- **Figure 5** compares training/test distributions but omits the **ind** dataset’s analysis in Appendix E, leaving readers uncertain about its stability.
- **Suggestion**: Release synthetic data with similar statistical properties (e.g., covariance matrices) or anonymized root-cause descriptions to enable deeper analysis without compromising confidentiality.

### **Evaluation Protocol and Metric Transparency**
The ALARM score (Appendix D) is central to the evaluation but is inadequately explained in the main text. The paper does not compare ALARM to established metrics (e.g., F1-score, precision-recall) or justify its superiority for operational consistency. For example:
- **Table 2** reports ALARM scores but does not contextualize how early detection (EDF) and false alarms (AAF) trade off against each other.
- **Suggestion**: Include a concise mathematical definition of ALARM in Section 4.1 and compare it with existing metrics using ablation studies (e.g., as done in [Gim and Min, 2025](https://doi.org/10.48550/arXiv.2305.09691)).

### **Baseline Methodology Limitations**
The baselines (mean-deviation detectors) are overly simplistic and fail to reflect modern TSAD approaches. For instance:
- **Table 2** shows suboptimal performance (e.g., ALARM=0.5 for **wat**), but recent methods like LSTM-based models [Hundman et al., 2018](https://arxiv.org/abs/1802.04431) or VAEs [Correia12 et al., 2023](https://arxiv.org/html/2309.02253v2) could achieve higher scores.
- **Suggestion**: Evaluate state-of-the-art models (e.g., methods from PyOD [Zhao et al., 2020](http://jmlr.org/papers/v20/19-011.html)) as baselines and report hyperparameter tuning protocols to ensure fair comparisons.

### **Temporal Leakage in Dataset Splits**
While buffer zones between training and test sets were removed (__Section 3.2__), the absence of detailed split strategies (e.g., windowing, stride) raises concerns about temporal leakage. For example:
- **Figure 4** shows most anomalies are short (<100 steps), but long-range dependencies might still exist.
- **Suggestion**: Explicitly describe split methodologies (e.g., sliding-window validation) and analyze temporal correlations in Appendix E to validate independence between splits.

### **Lack of Computational Efficiency Analysis**
The paper focuses on detection performance but omits computational metrics (e.g., inference speed, memory usage). For industrial deployment, these are critical:
- Appendix E lists hardware specs but does not report runtime or scalability benchmarks.
- **Suggestion**: Add efficiency metrics (e.g., FPS, FLOPS) for baseline models and discuss trade-offs between accuracy and latency.

_All references are cited in the original paper and can be found in the bibliography section._

**Strengths Contributions:**

- The core strength lies in introducing the **first publicly available real-world datasets** for chemical process anomaly detection, addressing the longstanding reliance on synthetic benchmarks like TEP [8]. The six datasets span **laboratory-scale, pilot-scale, and industrial-scale operations**, providing unprecedented access to multivariate sensor data with detailed anomaly annotations. Unlike prior benchmarks (SWaT [11], SMD [28]), these datasets capture authentic process complexity, including multi-phase anomalies and diverse operational modes. The inclusion of an **industrial dataset** from a major chemical company—rare due to confidentiality constraints—offers invaluable insights into long-term operational variability and naturally occurring failures.

- The paper's **technical novelty** centers on the **ALARM score**, a metric emphasizing early detection, precision, and operational consistency for industrial safety. Unlike point-wise metrics in SMAP/MSL [14], ALARM incorporates temporal context, ensuring evaluations reflect real-world priorities. The datasets' complexity is validated through baseline experiments: simple methods achieve suboptimal performance (ALARM scores < 0.7), confirming non-triviality. This contrasts with existing benchmarks criticized for trivial anomalies or positional biases [32]. Figure 3 demonstrates no exploitable positional bias, while Figure 4 highlights realistic short-duration anomalies.

- The **potential impact** is substantial, bridging academic research and industrial needs to enable ML-driven monitoring systems preventing catastrophic failures. The datasets' diversity—from ternary distillation mixtures to reactive systems—ensures broad applicability, while actuator data and multimodal extensions [1] support future research in causal modeling. Unlike TEP's simulated faults, NoBOOM's anomalies arise from real-world failures or controlled injections informed by historical incidents [19], as detailed in Section 3.3.

- The **presentation is well-structured** with logical organization and effective contextualization through figures like industrial accident timelines (Figure 1) and sensor data from near-miss incidents (Figure 2). Dataset statistics (Table 1) and baseline results (Table 2) enhance clarity, though some technical details require cross-referencing appendices. While industrial dataset confidentiality limits transparency, the authors ensure reproducibility by releasing raw data and anonymized metadata where possible.

---

> ### Author Rebuttal · Authors · 2025-07-31
>
> Thank you for your constructive criticism and thoughtful feedback. Below, we address each of your points. These explanations and additions aim to make the content more apparent to the reader. Please let us know if your concerns are addressed and appropriately resolved or if we should add any details.
>
> # Anomaly Density
> Anomalies are generally not used during training; all designated training sets are strictly anomaly‑free, so models are never exposed to faults during training. Anomalies are rare in practice, yet generally diverse by nature. In practice, anomalies are rare but highly heterogeneous, and a benchmark must showcase a broad spectrum of fault types to test generalisation meaningfully. Our evaluation protocol was chosen because it targets the detection of diverse anomalies while leaving the number of true negatives virtually unchanged. The ind dataset comes from a continuously operating industrial‑scale plant; its statistics, including anomaly frequency, mirror genuine field conditions and thus let researchers assess performance in a realistic setting. The laboratory‑scale datasets indeed contain a higher fraction of anomalies, but this is intentional: their richer label information and diversity of fault modes reveal subtle biases in detection algorithms and highlight areas for improvement. The six datasets offer complementary viewpoints and enable a more nuanced assessment of time‑series anomaly‑detection methods.
>
> # Industry Dataset
> The missing metadata should not impede reproducibility, as the data was anonymized at creation and will not be changed again. Due to the high number of features, we could not include the full Figure 5 in the paper's main section. However, we will gladly include the analysis of Figure 5 for the industrial dataset in the Appendix for the camera-ready version. It shows no significant distributional shift, similar to the analysis of the other datasets.
> Due to our confidentiality agreement with our industry partner, we cannot provide information about the root causes of anomalies or any information that could leak sensitive information about the plant or the run process. This includes a simulation that is sufficiently similar to the real process. We judged that the rare opportunity to release real industrial data outweighed this limitation. Synthetic time-series data for chemical processes in general that are sufficiently different from those of the industrial process exist in the form of the TEP [Downs & Vogel].
> Our dataset presents the first real-world time-series anomaly detection dataset from a real industrial-sized process. The accompanying laboratory datasets for a detailed fault analysis include detailed phase and root‑cause labels, complementing them while respecting confidentiality constraints.
>
> # Evaluation Protocol
> The ALARM score used in this study is introduced in another paper explaining its differences from other metrics and detailing its capabilities. The score was selected for its focus on anomalies and its formal guarantees, which match the strict requirements of our setting, which we highlight in Section 4.1. For a better reader's understanding, we will include a reference to the original paper and a more detailed discussion in Appendix D for the camera-ready version. We provide more details in the rebuttal to reviewer RuUy.
>
> We will expand the details on the evaluation in Appendix E to include a detailed description of the evaluated methods and the parameter grid. We will add evaluations of more well-established shallow baselines, including k-means and EIF, to further illustrate the complexity of the proposed datasets. The main goal of this paper is to introduce a valuable collection of datasets for future benchmarking of time-series anomaly detection methods. Many modern solutions come with a significant number of hyperparameters, and evaluations have shown inconsistencies in the performance of many complex approaches. Therefore, we will include two of the most robust and well-established baselines for evaluations [Malhotra-1, Malhotra-2] and leave extensive benchmarking for future work. We will provide full details of the evaluation in the Appendix and will update Table 2 in the main section to summarize the results across the evaluation. The following table shows the ALARM scores of each baseline method on all datasets. We will add the full evaluation with all metrics to the camera-ready version.
> | Method     | bpw   | abm   | wat   | but   | ome   | ind   |
> |------------|-------|-------|-------|-------|-------|--------|
> | threshold  | 1.87  | 1.20  | 1.31  | 2.15  | 1.79  | 2.38   |
> | kmeans     | 14.27 | 6.92  | -0.5  | 6.87  | 1.36  | 7.09   |
> | EIF        | 5.0   | 6.51  | 2.26  | 6.86  | 1.0   | 1.0    |
> | LSTM-AE    | 11.07 | **9.39** | -0.5  | 3.99  | 2.75  | **14.06** |
> | LSTM-P     | 8.13  | 3.40  | -0.5  | 1.96  | 4.0   | **8.42**  |
> | best       | 63.25 | 12.45 | 19.36 | 23.54 | 11.00 | 360.48 |
> The evaluation with shallow methods and comparatively simple deep baselines illustrates the complexity of the dataset and the need for more sophisticated methods to solve such complex tasks.
>
> [Downs & Vogel] Downs, James J., and Ernest F. Vogel. "A plant-wide industrial process control problem." Computers & chemical engineering 17.3 (1993): 245-255.
>
> [Malhotra-1] Malhotra, Pankaj, et al. "LSTM-based encoder-decoder for multi-sensor anomaly detection." arXiv preprint arXiv:1607.00148 (2016).
>
> [Malhotra-2] Malhotra, Pankaj, et al. "Long short term memory networks for anomaly detection in time series." Proceedings. Vol. 89. No. 9. 2015.

---

> > ### Comment · Reviewer_77qs · 2025-08-04
> >
> > First and foremost, sorry for the chunked feedback question in my initial response; it was in markdown format but didn't render correctly (I should double-check the comment after posting)
> > The initial question was
> > > Could you clarify the **split strategies** used for temporal leakage mitigation (e.g., windowing, stride)? How were long-range dependencies addressed?
> >
> > Besides, thank you for your detailed response and for agreeing to run additional experiments. Your clarifications on the intentional anomaly density and the understandable confidentiality constraints of the industrial dataset have resolved my primary concerns on those fronts.
> >
> > However, I believe the paper's contribution could be significantly strengthened by addressing two remaining issues regarding the Evaluation Protocol:
> >
> > - __Scope and Timeliness of Baselines__: Thank you for adding more baselines; the new results table powerfully demonstrates the datasets' complexity. However, labeling a thorough evaluation of modern methods as "future work" significantly curtails the immediate impact of a benchmark paper. The proposed LSTM baselines from Malhotra et al. are foundational but nearly a decade old.

---

> > > ### Author Response · Authors · 2025-08-05
> > > **Addressing remaining concerns**
> > >
> > > Thank you for your response. We are glad that we could alleviate most of your concerns with our initial response, and we will gladly elaborate on your two remaining concerns below.
> > >
> > > ## Benchmarking
> > > ### Shallow methods
> > > In this paper, **we introduce six novel datasets** collected from **real chemical processes** for anomaly detection in multivariate time series. Benchmarking is not the primary objective of this study. The main objective is to introduce and analyze the data to investigate its suitability for evaluations of time-series anomaly detection methods. For each dataset, we provide **detailed labels**, including anomaly phases, with additional details on anomaly causes for the laboratory-scale plants. Additionally, we provide a detailed analysis of potential artifacts that could interact with evaluations. It has been shown that simple and, in particular, shallow baselines can achieve competitive performance on many time-series anomaly detection datasets [1]. Such results question the true complexity of many datasets. Therefore, we focus on investigating the complexity of the proposed datasets by evaluating a suite of shallow baselines. The experiments show that these baselines achieve suboptimal performance across all datasets, suggesting that the proposed datasets are sufficiently complex to provide a significant challenge.
> > >
> > > ### Deep methods
> > > The true performance of time-series anomaly detection methods is generally opaque. The datasets evaluated often contain excessive distributional shift, questionable labels, or other artifacts that non-trivially interact with the evaluation protocol [1, 2]. Some evaluations use point-adjustment [3], which has been shown to inflate scores and, in particular, overestimate the performance of random predictions [4]. Other common metrics that avoid point-adjustment, e.g., point-wise metrics, have difficulty ranking methods intuitively, because they are unable to distinguish between prediction patterns in anomaly windows [2]. Consequently, it is not clear which methods should be selected to represent state-of-the-art performance. Furthermore, most evaluations are performed in specific settings where architecture and hyperparameters are specifically adapted. Due to the high variance of performance and hyperparameters [2], substantial adaptation and hyperparameter-tuning are fundamental for thorough benchmarking in novel settings. Considering the complexity of the most recent methods, such an evaluation is beyond the scope of this study.
> > >
> > > The methods we selected are comparatively simple, but also among the most consistently robust-performing methods across past benchmarks [2, 5, 6]. Therefore, these evaluations should provide strong baselines for future evaluations on the NoBOOM datasets.
> > >
> > > ## Temporal Leakage
> > > The process manager of the industrial plant provided suggestions for reasonable time frames in which most long-term effects should be resolved. These initial estimates were checked by other in-house domain experts, including the plant operator. We extended these suggestions generously by doubling the initial estimate. We will add these details of the splitting process to the camera-ready version to increase transparency. Due to our confidentiality agreement, we are not able to reveal the exact numbers.
> > >
> > > [1] https://www.dropbox.com/scl/fi/cwduv5idkwx9ci328nfpy/Problems-with-Time-Series-Anomaly-Detection.pdf?rlkey=d9mnqw4tuayyjsplu0u1t7ugg&e=1&dl=0
> > >
> > > [2] Wagner, Dennis, et al. "Timesead: Benchmarking deep multivariate time-series anomaly detection." Transactions on Machine Learning Research (2023).
> > >
> > > [3] Xu, Haowen, et al. "Unsupervised anomaly detection via variational auto-encoder for seasonal kpis in web applications." Proceedings of the 2018 world wide web conference. 2018.
> > >
> > > [4] Kim, Siwon, et al. "Towards a rigorous evaluation of time-series anomaly detection." Proceedings of the AAAI conference on artificial intelligence. Vol. 36. No. 7. 2022.
> > >
> > > [5] Schmidl, Sebastian, Phillip Wenig, and Thorsten Papenbrock. "Anomaly detection in time series: a comprehensive evaluation." Proceedings of the VLDB Endowment 15.9 (2022): 1779-1797.
> > >
> > > [6] Hartung, Fabian, et al. "Deep anomaly detection on tennessee eastman process data." Chemie Ingenieur Technik 95.7 (2023): 1077-1082.

---

> > > > ### Comment · Reviewer_77qs · 2025-08-05
> > > >
> > > > Thank you for the follow-up response. Most of my initial questions, including the new feedback question, are addressed. I am still conservative about the benchmarking with baselines, but overall, it turned out to be a good dataset for the scientific domain. I will revise my initial rating accordingly.

---

### Comment · Area_Chair_bjy4 · 2025-08-08

Dear Reviewers,

First, I want to appreciate your help in contributing NeurIPS review process. According to this year's policy, we may flag reviewers without any interaction with the authors -- so please let the authors and us know if they addressed your concerns.

Appreciate your help again!

---

### Note · Authors · 2025-08-15

We thank the reviewers for their constructive feedback.

This study introduces six novel datasets for time-series anomaly detection from real-life chemical plants, including one from a running industrial process. To illustrate the suitability of the datasets, we analyze their statistics including anomaly density, anomaly lengths, distributional shift, and general statistics. To illustrate the complexity of the tasks presented by these datasets, we evaluate a suite of shallow baselines and two simple deep baselines, which have been shown to solve related tasks. The analysis suggests a considerable complexity. Additionally, the datasets provide detailed labels of the phases of anomalies to enable more precise evaluations. We introduce two novel metrics that explicitly evaluate the responsiveness of an algorithm with respect to the phases of anomalies.

All reviewers have highlighted the potential impact of the proposed datasets and the complexity analysis as major strengths of this study.

We have addressed the concerns of the reviewers with the following changes:
- Added more shallow baselines and two robust deep baselines to strengthen the complexity analysis.
- Added detailed explanations of the proposed metrics in the appendix and improved their descriptions in the main paper to better highlight their intuitions.
- Added more details to the split strategy for the industrial process.
- Added the analysis of distributional shift on the industry dataset to the appendix, and added a summary and reference to the discussion in the main paper.
- Added more details on the evaluation protocol in Appendix E.

---

### Decision · Program_Chairs · 2025-09-18

**Decision:**

Accept (poster)

**Comment:**

This paper present NoBOOM - - a six real world datasets for time series AD (some are from lab and one is industrial). It includes phase labels and eval protocol centered on alarm.

Table 1 shows realistic scale and reviewers agree on this as substantial merit for the field.

Overall feedback is positive. Some initial questions include stronger, SOTA baselines, more details on data splits, and efficiency reporting. During rebuttal, the author added some shallow and deep ones and describe a conservative split strategy.

Overall, I believe this is solid contribution to industrial AD.